# Structure–function analysis of the extracellular domain of the pneumococcal cell division site positioning protein MapZ

Sylvie Manuse[1,2,*], Nicolas L. Jean[3,4,5,*], Mégane Guinot[1,2], Jean-Pierre Lavergne[1,2], Cédric Laguri[3,4,5], Catherine M. Bougault[3,4,5], Michael S. VanNieuwenhze[6], Christophe Grangeasse[1,2,**] & Jean-Pierre Simorre[3,4,5,**]

Accurate placement of the bacterial division site is a prerequisite for the generation of two viable and identical daughter cells. In *Streptococcus pneumoniae*, the positive regulatory mechanism involving the membrane protein MapZ positions precisely the conserved cell division protein FtsZ at the cell centre. Here we characterize the structure of the extracellular domain of MapZ and show that it displays a bi-modular structure composed of two subdomains separated by a flexible serine-rich linker. We further demonstrate *in vivo* that the N-terminal subdomain serves as a pedestal for the C-terminal subdomain, which determines the ability of MapZ to mark the division site. The C-terminal subdomain displays a patch of conserved amino acids and we show that this patch defines a structural motif crucial for MapZ function. Altogether, this structure–function analysis of MapZ provides the first molecular characterization of a positive regulatory process of bacterial cell division.

[1] CNRS, Molecular Microbiology and Structural Biochemistry, UMR 5086, 7 passage du Vercors, Lyon 69367, France. [2] Université Lyon 1, Molecular Microbiology and Structural Biochemistry, UMR 5086, 7 passage du Vercors, Lyon 69367, France. [3] Université Grenoble Alpes, Institut de Biologie Structurale, 71 avenue des Martyrs—CS10090, Grenoble cedex 9 38044, France. [4] CEA, DSV, Institut de Biologie Structurale, 71 avenue des Martyrs—CS10090, Grenoble cedex 9 38044, France. [5] CNRS, Institut de Biologie Structurale, 71 avenue des Martyrs—CS10090, Grenoble cedex 9 38044, France. [6] Department of Chemistry, Indiana University, 800 E. Kirkwood Avenue, Bloomington, Indiana 47405-7102, USA. * These authors contributed equally to this work. ** These authors jointly supervised this work. Correspondence and requests for materials should be addressed to C.G. (email: c.grangeasse@ibcp.fr) or to J.-P.S. (email: jean-pierre.simorre@ibs.fr).

In bacteria, binary fission is the most common method of cell division[1]. For that, bacteria have evolved different mechanisms to select the division site and to position the tubulin-like protein FtsZ at mid-cell[2]. FtsZ forms a contractile ring called the Z-ring that encircles the medial portion of the cell. Robust models for positioning the Z-ring at mid-cell have emerged in model bacteria such as *Escherichia coli* and *Bacillus subtilis*. Indeed, two main systems, the nucleoid occlusion and the Min system, both prevent the Z-ring to assemble anywhere in the cell other than at mid-cell[3]. However, some recent data show that these two systems are not alone at play in *E. coli* and *B. subtilis*, and that other mechanisms, not well identified yet, should contribute to determine mid-cell very early in the cell cycle[4]. Many bacteria also lack both of these systems[5]. This is well described in the bacterium *Caulobacter crescentus*, in which Z-ring positioning is governed by the MipZ protein[6]. Very recently, two systems, the PomZ and the SsgA/SsgB systems, at odds with the nucleoid occlusion, Min and MipZ systems, have been identified in *Myxococcus xanthus*[7] and *Streptomyces coelicolor*[8], respectively. These two systems do not act negatively to prevent Z-ring positioning and assembly anywhere other than at mid-cell. Instead, they act positively to promote FtsZ assembly at mid-cell. However, it still remains unclear how these two systems themselves identify and position at mid-cell.

Interestingly, some bacteria are devoid of any of the systems described so far and how they identify their mid-cell as well as how they position the divisome machinery, remain unknown. This is the case of the bacterium *Streptococcus pneumoniae*, a Gram-positive bacterium that is a frequent cause of community-acquired diseases[9]. In this species, an unprecedented positive mechanism of cell division has been shown to allow selection of the division site and positioning of the divisome at mid-cell[10,11]. This system relies on the newly identified protein MapZ and the serine/threonine-kinase StkP, a crucial regulator of pneumococcal cell division and morphogenesis[12–14]. MapZ is conserved in *streptococci*, *lactococci* and most *enterococci*, suggesting that this positive regulatory mechanism is widespread in many bacterial species[10]. This protein, which is formed by a cytoplasmic and an extracellular domain linked together by a single transmembrane span, localizes as a ring at mid-cell in newborn cells[10,11]. As the cell elongates, the MapZ ring splits in two new rings that move apart by the synthesis of peptidoglycan that occurs at mid-cell and that elongates the cell. On completion of cell elongation, and synthesis of the new cell halves, the two MapZ rings are thus positioned at the future division site (the cell equator) of the two daughter cells, while FtsZ still localizes at the constricting septum (the division site)[10]. MapZ therefore precedes FtsZ at the cell equator. Then, as cell constriction proceeds, FtsZ also localizes at the cell equator at a place that coincides with that of MapZ. Together with the finding that FtsZ interacts with MapZ, the latter protein is proposed to act as a molecular beacon of the cell equator that further positions precisely FtsZ at mid-cell[10]. It was demonstrated that MapZ phosphorylation by StkP does not affect this function of MapZ. However, and strikingly, while the two rings of MapZ move towards the cell equator, a third ring of MapZ localizes at the constricting septum and persists together with FtsZ till the end of cell constriction[10]. There, two threonines of the cytoplasmic domain of MapZ are phosphorylated by StkP. This phosphorylation controls the integrity, as well as the closure of the Z-ring during cell constriction. Indeed, when MapZ phosphorylation is deregulated, the Z-ring closes faster and hinders chromosome segregation, and its integrity is compromised[10]. Very interestingly, the phosphorylation of MapZ influences neither its binding to FtsZ, nor the GTPase FtsZ activity, nor the polymerization of FtsZ, suggesting that MapZ phosphorylation influences indirectly the constriction of the

Z-ring[10]. MapZ would thus act as a dual-function protein, which not only serves to mark the division site and position FtsZ but that also controls the Z-ring closure[10]. This mechanism is at odds with the regulatory systems controlling the Z-ring positioning and closure in other bacterial models studied so far.

In our ongoing efforts to characterize this mechanism and decipher how MapZ positions at the cell equator, we have performed a structure–function study of the extracellular domain of MapZ (MapZ$_{extra}$). In this work, we report the nuclear magnetic resonance (NMR) structure of MapZ$_{extra}$ and show that it displays a new bi-modular structure composed of two subdomains separated by a flexible serine-rich linker. Next, we have analysed the importance of these two subdomains in pneumococcal cell division. Our study shows that a conserved patch of amino acids in the C-terminal domain plays a crucial function in binding peptidoglycan and positioning MapZ at the cell equator, whereas the N-terminal domain behaves as a pedestal indispensable for the function of the C-terminal domain. Altogether, this work represents the first molecular characterization of the MapZ system.

## Results

**MapZ extracellular domain is divided in two subdomains**. The extracellular domain of MapZ (MapZ$_{extra}$) stretches from residues 182 to 464 (Fig. 1a). Preliminary amino-acid sequence analysis showed the presence of a serine-rich region (residues 314–354) predicted to be disordered by the IUPred server[15] and to contain poorly conserved amino acids (Supplementary Fig. 1). This region is flanked by two polypeptides containing more conserved and possibly more structured amino acids. In the absence of reliable structure prediction for MapZ$_{extra}$, we first produced and purified $^{15}$N- and/or $^{13}$C-labelled MapZ$_{extra}$, as well as the two prospective subdomains, MapZ$_{extra1}$ (residues 182–313) and MapZ$_{extra2}$ (residues 355–464; Supplementary Fig. 2A). Two-dimensional (2D)-[$^{1}$H,$^{15}$N] correlation NMR spectra were recorded on both subdomains, which display signal dispersion patterns characteristic of two folded and stable domains (Fig. 1b). The same procedure was applied to the full-length $^{15}$N-labelled MapZ$_{extra}$ domain. Its 2D-[$^{1}$H,$^{15}$N] correlation NMR spectrum displayed 258 peaks out of the 278 peaks expected (Fig. 1b). The characteristic signatures of the two individually structured subdomains, MapZ$_{extra1}$ and MapZ$_{extra2}$, were perfectly identified in the MapZ$_{extra}$ spectrum, confirming the presence of two stable subdomains in MapZ$_{extra}$ (Fig. 1c; Supplementary Fig. 3A). These results suggest that the two subdomains can fold independently and that their structure can be determined separately.

**Motional independence of the two subdomains of MapZ$_{extra}$**. The two MapZ$_{extra1}$ and MapZ$_{extra2}$ subdomains are linked by a long polypeptide containing a serine-rich region (serine-rich linker, SRL) from residues K314 to S354. Whereas the majority of the amide protons of this SRL were not detected in the 2D-[$^{1}$H,$^{15}$N] correlation spectrum recorded at pH 7.5, intense resonances appear at pH 4.5 with chemical shifts that were characteristic of a disordered structure (8.1–8.5 p.p.m.; Supplementary Fig. 3B). The decrease of the amide proton exchange rates at low pH leading to the resonance detection and the weak dispersion of amide resonances are consistent with the IUPred prediction for the flexibility of the SRL (Supplementary Fig. 1). NMR relaxation measurements ($R_2/R_1$) were performed to analyse the relative rotational diffusion of the two subdomains in the full-length extracellular domain (Supplementary Fig. 4). In the absence of fast motion, $R_2/R_1$ values directly depend on the average rotational correlation time of the subdomains[16]. Average $R_2/R_1$ values measured for residues of MapZ$_{extra1}$ and MapZ$_{extra2}$

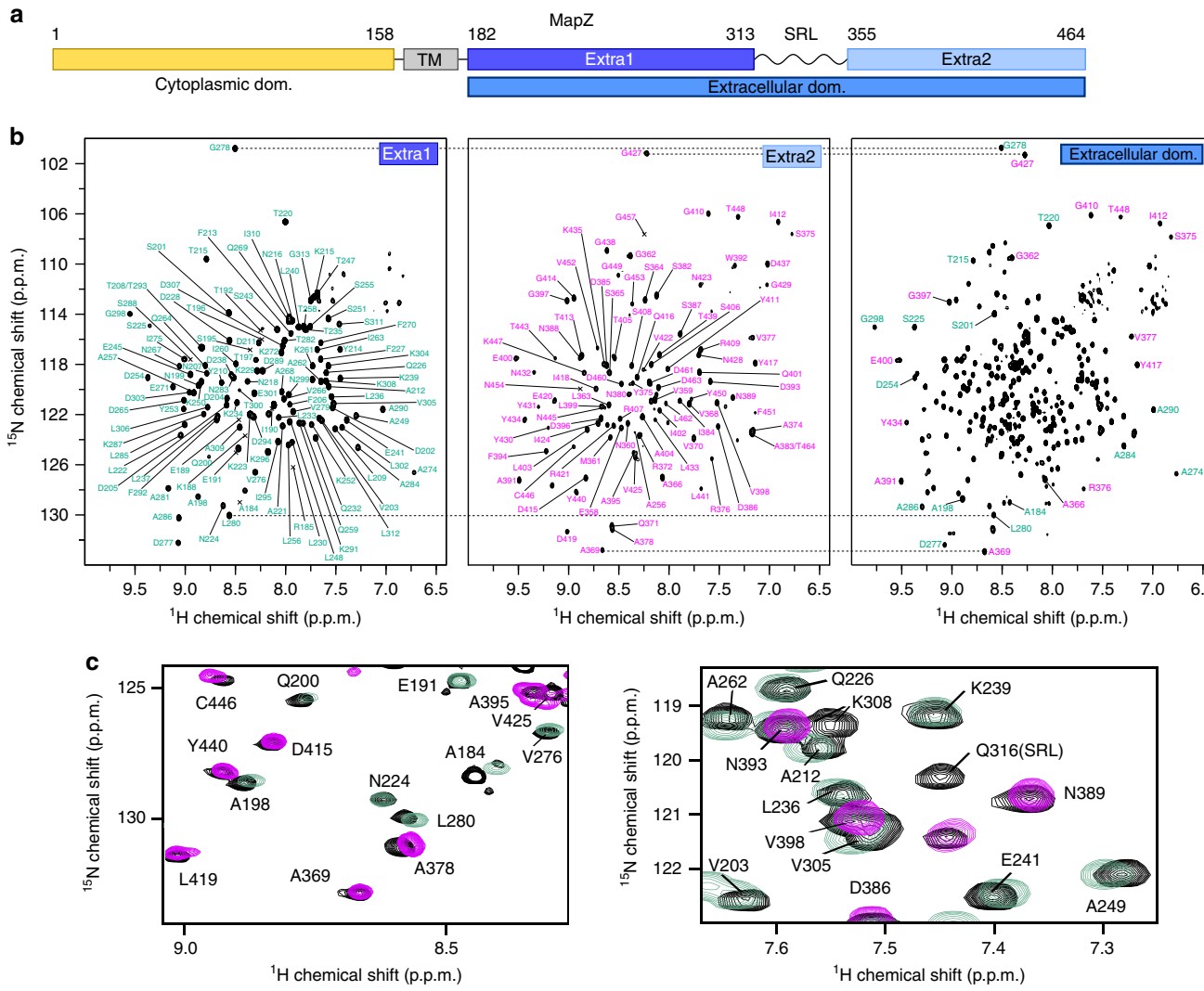

**Figure 1 | Organization of MapZ and NMR characterization of its extracellular domain.** (**a**) Architecture of MapZ. MapZ is composed of a cytoplasmic (in yellow) and an extracellular domain (in blue; residues 1–158 and 182–464, respectively), connected through a transmembrane α-helix (in grey). The extracellular domain of MapZ (MapZ$_{extra}$) is predicted by bioinformatic approaches[15] to be divided in two independent subdomains MapZ$_{extra1}$ (in dark blue) and MapZ$_{extra2}$ (in pale blue) linked by a flexible serine-rich stretch (from residues 314–354). MapZ$_{extra1}$ and MapZ$_{extra2}$ extend from residues 182–313 and 355–464, respectively. (**b**) 2D-[$^1$H,$^{15}$N]-BEST-TROSY of MapZ$_{extra1}$ (left), MapZ$_{extra2}$ (middle) and MapZ$_{extra}$ (right). Assignment of the resonances of the MapZ$_{extra1}$ and MapZ$_{extra2}$ subdomains is displayed in turquoise and magenta, respectively. A full resonance assignment of the MapZ$_{extra}$ spectrum is reported in Supplementary Fig. 3. (**c**) Excerpts of the superimposition of the 2D-[$^1$H,$^{15}$N]-BEST-TROSY spectra of MapZ$_{extra1}$ (turquoise), MapZ$_{extra2}$ (magenta) and MapZ$_{extra}$ (black). Assignments for the full-length construct are reported in black for each of the resonances. They all show a good overlay with resonances of the individual subdomains, with the exception of K308 that is affected due to its location at the C terminus of MapZ$_{extra1}$ (in the later construct the resonance is superimposed with the resonance in magenta of N393) and of Q316, which is part of the SRL and thus absent from MapZ$_{extra1}$ or MapZ$_{extra2}$. This superimposition highlights the preservation of the fold of the isolated subdomains in the full-length construct, as outlined by dashed lines in **b**.

subdomains were 14.24 and 7.23, respectively. This large difference in the $R_2/R_1$ ratios is in favour of an independent hydrodynamic behaviour of the two subdomains, thus suggesting the absence of a strong interaction between them. The independent mobility of the two domains conferred by the flexible SRL was confirmed by small-angle X-ray scattering (SAXS) data recorded on the full-length extracellular domain at two concentrations (Supplementary Fig. 5A). Indeed, the Kratky plot derived from the experimental $I(s)$ scattering curves showed a behaviour for $s > 2\,nm^{-1}$ in agreement with the presence of a large flexibility (Supplementary Fig. 5B). Surface plasmon resonance recorded with the two individual subdomains further confirmed that the MapZ$_{extra1}$ and MapZ$_{extra2}$ subdomains do not have any propensity to interact (Supplementary Fig. 5C).

This result is corroborated by the absence of chemical shift variations measured on the 2D-[$^1$H,$^{15}$N] correlation NMR spectrum recorded on MapZ$_{extra1}$ before and after addition of MapZ$_{extra2}$ at a ratio of 1:2 (Supplementary Fig. 5D). All these results point out a bi-modular organization of the extracellular domain of MapZ, with two independent subdomains connected by a highly dynamic SRL.

**Solution structures of the two subdomains of MapZ$_{extra}$.** To calculate an ensemble of structures for the two MapZ extracellular subdomains, three-dimensional (3D)-NMR experiments were recorded on MapZ$_{extra1}$ and MapZ$_{extra2}$. $^1$H, $^{15}$N and $^{13}$C resonances were assigned with 86.9%, 87.3% and 93.8% completion in MapZ$_{extra1}$, and 89.8%, 87.4% and 91.0%

completion in MapZ$_{extra2}$, respectively A total of 202 (MapZ$_{extra1}$) and 160 (MapZ$_{extra2}$) phi/psi dihedral angles restraints were derived from the chemical shift analysis of backbone atoms. Furthermore, 3,602 and 3,167 distance constraints were extracted from 3D-edited nuclear Överhauser enhancement spectroscopy (NOESY) experiments for MapZ$_{extra1}$ and MapZ$_{extra2}$ subdomains, respectively. These data were used to generate 20 low-energy structures (Fig. 2) after eight iterations using Crystallography and NMR system (CNS) software[17] and Aria protocols[18]. Structural statistics for each of the two subdomains structure calculations are given in Table 1.

MapZ$_{extra1}$ is a 50-Å prolate domain (Fig. 2a) composed of four helices (H1: 200–213, H2: 225–240, H3: 245–269 and H4: 300–310). The H2 helix consists in a 3$_{10}$ helix (225–229) prolonged by an α-helix (230–240), thus inducing a bending of the helix axis. The N-terminal end of MapZ$_{extra1}$ is connected to the transmembrane segment of MapZ through a flexible linker, from residues 182 to 199, as suggested by {$^1$H}$^{15}$N-NOE and $R_2/R_1$ relaxation data (Supplementary Fig. 6A) and in agreement with the IUPred predictions (Supplementary Fig. 1). The H3 helix, which extends along the whole subdomain, is flanked by H1 and H2 at its bottom, and H4 at its top. The strong stability of the tertiary structure is ensured by the presence of numerous hydrophobic contacts between the different helices (Fig. 2c). The majority of the highly conserved residues are not accessible at the surface (Fig. 3), but are directly involved in many inter-helices interactions (Fig. 2c), yielding a very rigid domain. Searches for

structural homologues of the MapZ$_{extra1}$ subdomain prevented us to get further insight into its functional role[19]. Indeed, no MapZ$_{extra1}$ homologues were found in the Protein Data Bank (PDB) and a closest structural alignments were found almost exclusively with eukaryotic α-helix-rich proteins such as VPS4 (PDB 2LXL, Z-score 5.3, root mean squared deviation (r.m.s.d.) = 2.9 Å) and coatomer (PDB 3MKR, Z-score 5.0, r.m.s.d. = 3.5 Å).

Unlike MapZ$_{extra1}$, the MapZ$_{extra2}$ structure displays a globular fold (Fig. 2a). It is composed of a central four-stranded antiparallel beta-sheet (β1: 418–423, β2: 430–434, β3: 440–445 and β4: 449–452) surrounded by two α-helices (H5: 381–384 and H6: 398–409) and a small 3$_{10}$ helix (390–392; Fig. 2b). The subdomain ends with a partially flexible C terminus (454–464; Supplementary Fig. 6B). The electrostatic surface of MapZ$_{extra2}$ does not display any obvious positively or negatively charged patch (Supplementary Fig. 7). Conversely, amino-acid conservation obtained from multiple sequence alignment evidences a large patch at the surface of one of the faces (Fig. 3b). As pointed out in Fig. 3c, conserved residues involved in this patch are mainly hydrophilic in nature, suggesting that potential intermolecular interactions are mediated by hydrogen bonding, rather than electrostatic or hydrophobic networks. In addition, MapZ$_{extra2}$ does not have any close homologue in the PDB preventing us to hypothesize for a particular interactant. Indeed, the closest homologue, the alanyl-tRNA synthetase (PDB 2ZTG), only showed a Z-score value of 4 and an r.m.s.d. of 3.3 Å.

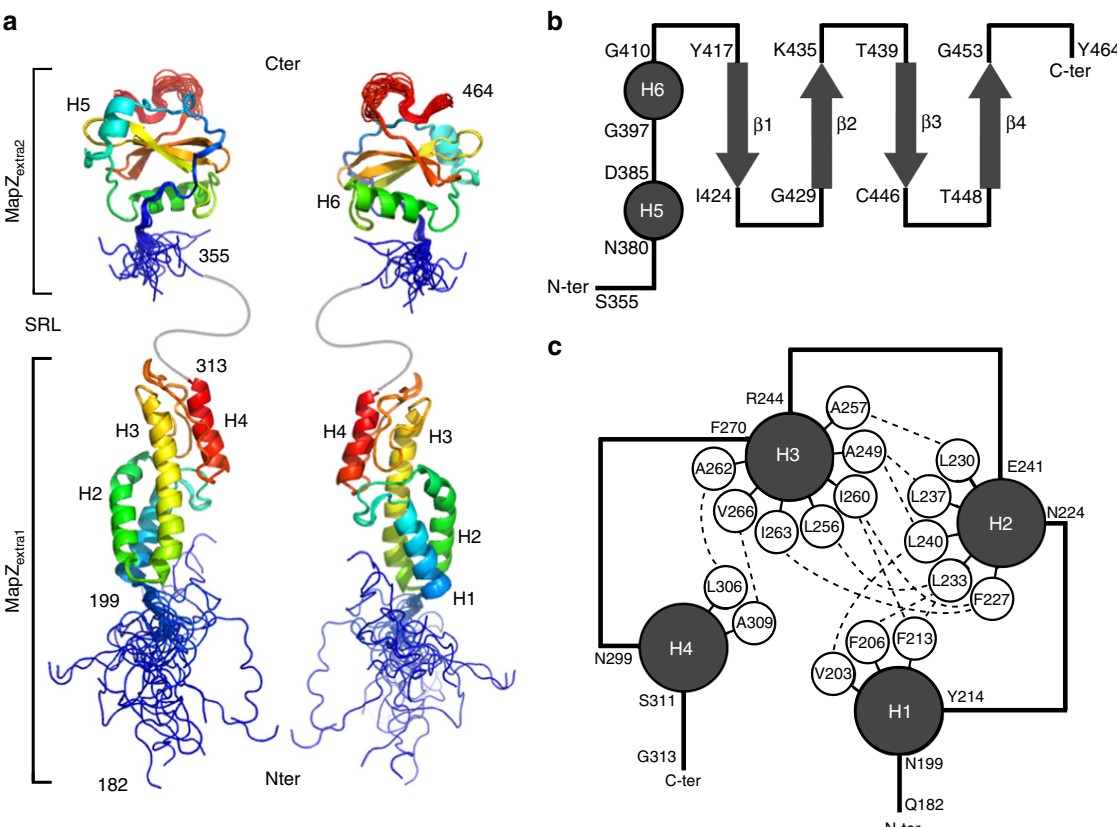

**Figure 2 | Structural analysis of MapZ$_{extra}$.** (**a**) Cartoon representation of the NMR ensemble of the 20 lowest-energy structures of MapZ$_{extra1}$ (lower structure) and MapZ$_{extra2}$ (upper structure). The ensemble on the right is rotated by 180° along the y axis compared with the structures on the left. The serine-rich linker (SRL) is pictured as a grey curved line. N- and C-terminal residues as well as α-helices are annotated. (**b**) Topology of the MapZ$_{extra2}$ subdomain. α-Helices and β-strands are represented by grey circles and arrows, respectively. (**c**) Topology of the MapZ$_{extra1}$ subdomain. Similarly to **b**, α-helices are shown as grey circles. The most significant residues that are implicated in hydrophobic interactions and maintain the global structure are displayed as white circles, while hydrophobic contacts are emphasized as dotted lines. In **b** and **c**, residues delimiting the loops pictured as black lines are indicated.

**Table 1 | Structural statistics for the ensemble of 20 NMR structures of MapZ$_{extra1}$ (PDB code 2ND9) and MapZ$_{extra2}$ (PDB code 2NDA).**

| | MapZ$_{extra1}$ | MapZ$_{extra2}$ |
|---|---|---|
| **NMR distance and dihedral constraints** | | |
| *Distance constraints* | | |
| Total unambiguous NOE restraints | 3,400 | 2,520 |
| Intra-residue | 1,033 | 903 |
| Inter-residue | 2,367 | 1,617 |
| Sequential ($|i - j| = 1$) | 668 | 468 |
| Medium range ($|i - j| \leq 5$) | 858 | 491 |
| Long range ($|i - j| > 5$) | 841 | 658 |
| Intermolecular | 0 | 0 |
| Total ambiguous NOE restraints | 832 | 647 |
| Hydrogen bonds | 0 | 0 |
| Total dihedral angle restraints | 202 | 160 |
| Phi | 101 | 80 |
| Psi | 101 | 80 |
| | | |
| Structure statistics[a] | | |
| *Violations (mean and s.d.)* | | |
| Distance constraints (Å) | 0.0759 ± 0.0053 | 0.0496 ± 0.0009 |
| Dihedral angle constraints (°) | 2.74 ± 0.0260 | 3.30 ± 0.0394 |
| Maximum dihedral angle violation (°) | 4.199 | 25.761 |
| Maximum distance constraint violation (Å) | 2.17 | 0.613 |
| *Deviations from idealized geometry* | | |
| Bond lengths (Å) | 0.0118 ± 0.0001 | 0.0074 ± 0.0001 |
| Bond angles (°) | 1.0142 ± 0.0106 | 0.7572 ± 0.0094 |
| Impropers (°) | 2.1571 ± 0.0481 | 1.8049 ± 0.038 |
| *Average pairwise r.m.s.d.[b] (Å)* | | |
| Heavy | 0.46 ± 0.06 | 0.40 ± 0.05 |
| Backbone | 0.13 ± 0.04 | 0.14 ± 0.03 |

NOE, nuclear Overhauser effect; r.m.s.d., root mean squared deviation.
[a]Pairwise deviations were calculated among 20 refined structures.
[b]These values were calculated on residues 199–312 for MapZ$_{extra1}$ and 362–453 for MapZ$_{extra2}$.

**MapZ$_{extra2}$ controls MapZ positioning at the cell equator**. As described above, the majority of conserved residues are hydrophobic and not solvent exposed in MapZ$_{extra1}$, whereas they are largely hydrophilic and form a large patch at the surface of MapZ$_{extra2}$. This suggested that MapZ$_{extra2}$ could be of crucial importance in the function of MapZ. We therefore first constructed a mutant in which the chromosomal copy of *mapZ* is substituted for *mapZ* deleted from the DNA fragment coding for MapZ$_{extra2}$ (strain *mapZΔextra2*) (Fig. 4a). The absence of the MapZ$_{extra2}$ domain is detrimental to the pneumococcus as indicated by the 31% increase in generation time and the 24.4% decrease in cell viability (Supplementary Table 1). To further evaluate the impact on the pneumococcal cell cycle, the mutant was observed microscopically (Fig. 4b). Most if not all *mapZΔextra2* cells displayed severe cell shape defects. More precisely, we mainly observed the presence of aberrant elongated cells and mini-cells together with some monster cells as previously detected for the MapZ null mutant[10]. Cell length and width measurements confirmed this visual impression, and showed that *mapZΔextra2* cells were significantly shorter and/or longer than wild-type (WT) cells (Supplementary Fig. 8).

To check whether these defects are due to the inability of MapZ to properly mark the division site when devoid of MapZ$_{extra2}$, we constructed the strain *gfp-mapZΔextra2* that expressed the green fluorescent protein (GFP) fused with the N-terminal end of MapZ devoid of the MapZ$_{extra2}$ domain (GFP-MapZΔextra2 fusion) as the single copy of *mapZ* from its native chromosomal locus under the control of the native promoter. As control, GFP-MapZΔextra2 was expressed at similar levels than GFP-MapZ in *gfp-mapZΔextra2* and WT strains, respectively (Supplementary Fig. 9). Strikingly, GFP-MapZΔextra2 did not adopt its characteristic three-ring localization pattern (two rings positioned at the two future division sites (cell equators) and flanking a third ring at the constricting septum (division site in WT cells)) (Fig. 5)[10]. Rather, we observed that GFP-MapZΔextra2 delocalized around the cell in the membrane, while a single ring persisted at the constricting septum. This shows that MapZ$_{extra2}$ is required for the positioning of MapZ at the cell equators to mark the future division site. Supporting this, the localization of FtsZ fused with the GFP[14] in *mapZΔextra2* cells was altered and the structure of the Z-ring appeared compromised (Supplementary Fig. 10). In addition, this also shows that MapZ$_{extra2}$ is likely not required for MapZ positioning at the constricting septum.

**A conserved patch of amino-acid positions MapZ at mid-cell**. To further characterize the importance of MapZ$_{extra2}$, we performed a series of combined mutations of several of the conserved and surface-exposed amino acids (Fig. 3). Before transforming these mutations in pneumococcal cells, we first cloned and overproduced a series of MapZ$_{extra2}$ mutants to analyse whether their fold remained similar to the structure of WT MapZ$_{extra2}$. Unfortunately, individual or combined mutations of S375, R376, Y379, N380, D420, R421, N445, T448 and D463 led to protein denaturation, compromising their subsequent purification and a relevant analysis of mutation impact in *S. pneumoniae* cells. However, we succeeded in mutating up to seven conserved amino acids at a time (R409A, Y411A, N428A, Y430F, Y450A, F451L and N454A) without impacting MapZ$_{extra2}$ folding (Supplementary Fig. 2B). These amino acids are adjacent at the surface of MapZ$_{extra2}$ forming a

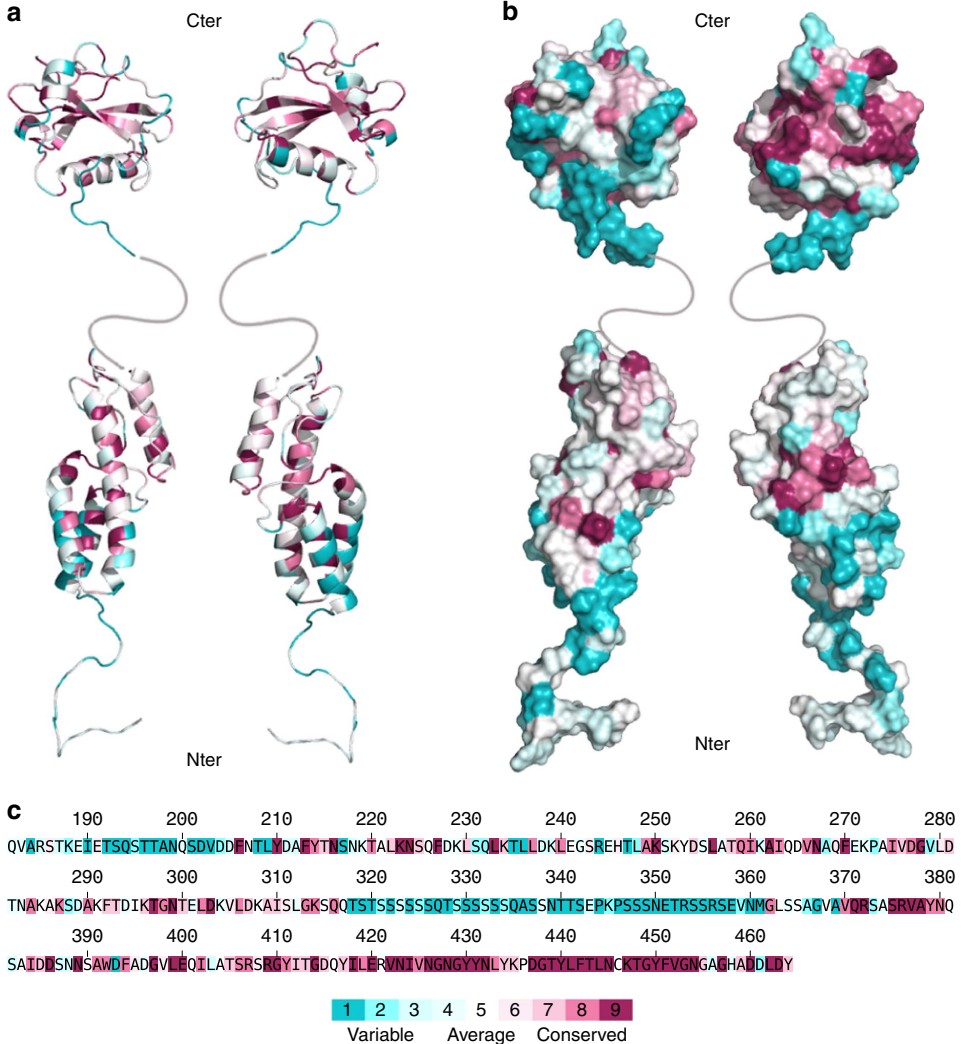

**Figure 3 | Amino-acid conservation in MapZ_extra.** Sequence conservation scores (**a**) inside or (**b**) at the surface of MapZ_extra NMR structures was calculated by the Consurf webserver[50]. Scores range from 1 (not conserved, cyan) to 9 (highly conserved, magenta). In **a** and **b**, left and right structures are rotated by 180° along the y axis. These scores are also displayed on the MapZ_extra sequence (**c**), with the Consurf colour code reported below.

conserved long polar cluster (Supplementary Fig. 11). We therefore constructed a pneumococcal strain in which the seven surface-exposed and -conserved amino acids were mutated (Fig. 4a). The resulting *mapZ-extra2Mut* strain showed cell growth, viability and shape defects similar to those observed for the *mapZΔextra2* strain (Fig. 4b; Supplementary Table 1; Supplementary Fig. 8). When analysing the localization of GFP-MapZ_extra2Mut, we also observed that it was largely impaired. Indeed, the two outer-ring of MapZ delocalized in the membrane and only the MapZ ring persisted at the constricting septum (Fig. 5). We checked that the GFP-MapZ_extra2Mut fusion was produced as in WT cells (Supplementary Fig. 9). This shows that the conserved surface delineated by at least seven amino acids is crucial for the positioning of MapZ at the cell equator.

The inability of GFP-MapZ_extra2Mut to position properly at the cell equator and to mark the division site suggested that MapZ_extra2Mut could not be shifted with the peptidoglycan produced at the constricting septum and required for cell elongation[5,10]. To test this, we first performed a pull-down assay to assess the ability of MapZ_extra2 to bind peptidoglycan. As shown in Fig. 6a and Supplementary Fig. 12, MapZ_extra2 was efficiently pulled down by purified cell wall sacculi of

*S. pneumoniae*. By contrast, cell wall binding to MapZ_extra2Mut was strongly reduced. In a second experiment, we tracked peptidoglycan synthesis together with the localization of GFP-MapZ_extra2Mut *in vivo* (Fig. 6b). For that, *gfp-mapZ-extra2Mut* cells were sequentially incubated in the presence of the two fluorescently labelled D-amino acids, TDL and HADA, that are specifically incorporated into peptidoglycan during cell elongation[10,20]. As control, WT cells were treated using the same procedure and as previously described, the most recent synthesized peptidoglycan at mid-cell (Fig. 6b, blue labelling, second pulse) shifted the previously incorporated one (Fig. 6b, red labelling, first pulse) and both were flanked by the two outer rings of MapZ that mark the future division sites. However, this localization pattern was not observed in *gfp-mapZ-extra2Mut* cells (Fig. 6b). While peptidoglycan synthesis occurred properly at the constricting septum, GFP-MapZ_extra2Mut did not form two rings at the cell equator. Last but importantly, MapZ is unable to bind peptidoglycan sacculi from *E. coli* or *B. subtilis* (Supplementary Fig. 12). Altogether, these data show that the seven conserved amino acids in MapZ_extra2 are required for MapZ-specific binding to pneumococcal peptidoglycan and thus for MapZ-mediated beaconing of the division site.

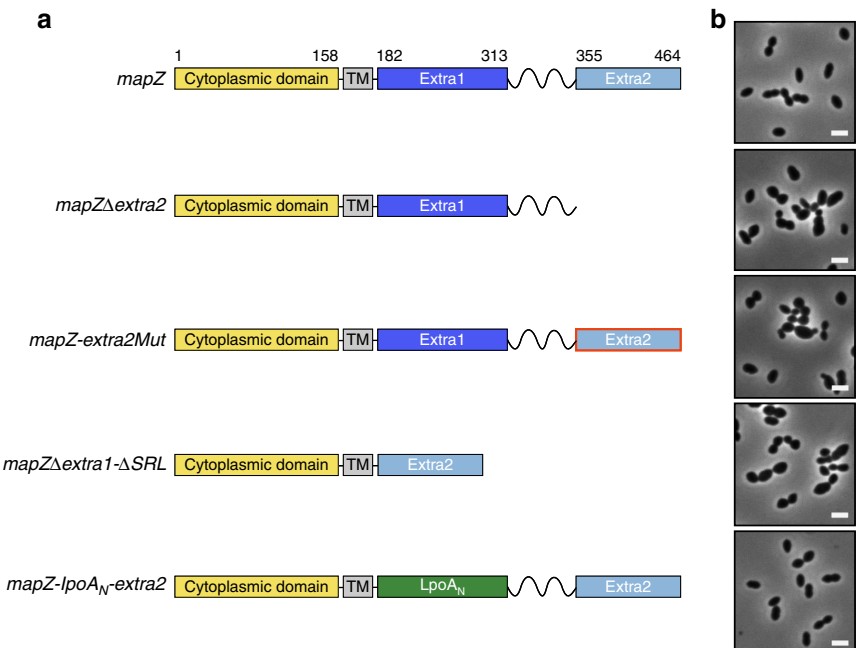

**Figure 4 | Schematic organization of MapZ mutants and impact on the cell shape.** (**a**) Schematic topological organization of the proteins coded by *mapZ* mutants. The cytoplasmic domain and the transmembrane domain of MapZ are shown in yellow and dark grey, respectively. The poly-serine linker connecting the two subdomains MapZ$_{extra1}$ (in dark blue) and MapZ$_{extra2}$ (in pale blue) is shown as a zig-zag line. The green box represents substitution of MapZ$_{extra1}$ for the LpoA$_N$ domain of *E. coli*. The red edge around MapZ$_{extra2}$ symbolizes the seven mutations of conserved amino acids. (**b**) Phase-contrast microscopy images of exponentially growing cells harbouring the wild-type *mapZ*, *mapZΔextra2*, *mapZ-extra2Mut*, *mapZΔextra1-ΔSRL* and *mapZ-lpoA$_N$-extra2* genes at 37 °C in Todd Hewitt Yeast medium. Scale bar, 2 μm.

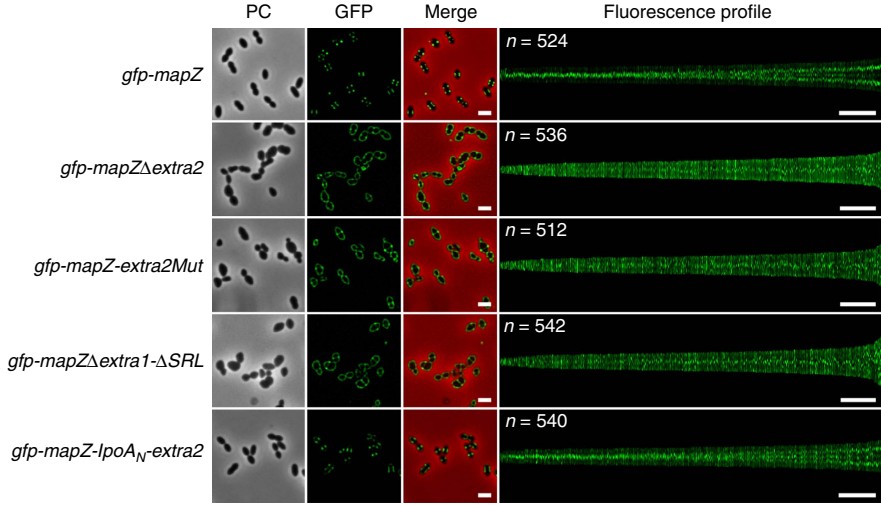

**Figure 5 | Localization of MapZ mutated forms in *mapZ* mutants.** GFP-MapZ localization in wild-type *gfp-mapZ*, *gfp-mapZΔextra2*, *gfp-mapZ-extra2Mut*, *gfp-mapZΔextra1-ΔSRL* and *gfp-mapZ-lpoA$_N$-extra2* strains. Phase contrast (left, PC), GFP fluorescence signal (middle, GFP) and overlays (right, Merge) between phase-contrast (red) and GFP (green) images are shown. The maps of fluorescence profile of cells sorted according to their length are presented in the far right column for wild-type and *mapZ* mutant cells. The total integrated fluorescence intensity of each cell (*y* axis) is plotted as a function of its cell length (*x* axis). Cells are sorted according to increasing cell length from left to right on the later axis. For each fluorescence profile, *n* indicates the total number of cells analysed. Scale bars on the microscopy images and fluorescence profiles correspond to 2 and 3 μm, respectively.

**MapZ$_{extra1}$ and the SRL act as a pedestal for MapZ$_{extra2}$.** The rigid four α-helix bundle of MapZ$_{extra1}$ is flanked by two flexible linkers: the flexible linker connecting the transmembrane domain to the N-terminal end of the bundle and the SRL separating MapZ$_{extra1}$ from MapZ$_{extra2}$. We envisioned that this particular modular organization could contribute to localize MapZ$_{extra2}$ in a particular region of the cell wall and to allow MapZ moving towards the cell equator during cell elongation. To test this, we constructed a mutant expressing *mapZ* deleted of both *mapZ$_{extra1}$* and the region coding for the SRL (strain *mapZΔextra1-ΔSRL*; Fig. 4a). Strikingly, *mapZΔextra1-ΔSRL* cell viability and generation time were severely impacted and comparable to that of *mapZΔextra2* (Supplementary Table 1). The same was true for cell morphology. Indeed, nearly all *mapZΔextra1-ΔSRL* cells displayed cell shape defects as reflected by the presence of aberrant-shaped cells and mini-cells (Fig. 4b; Supplementary Fig. 8). When

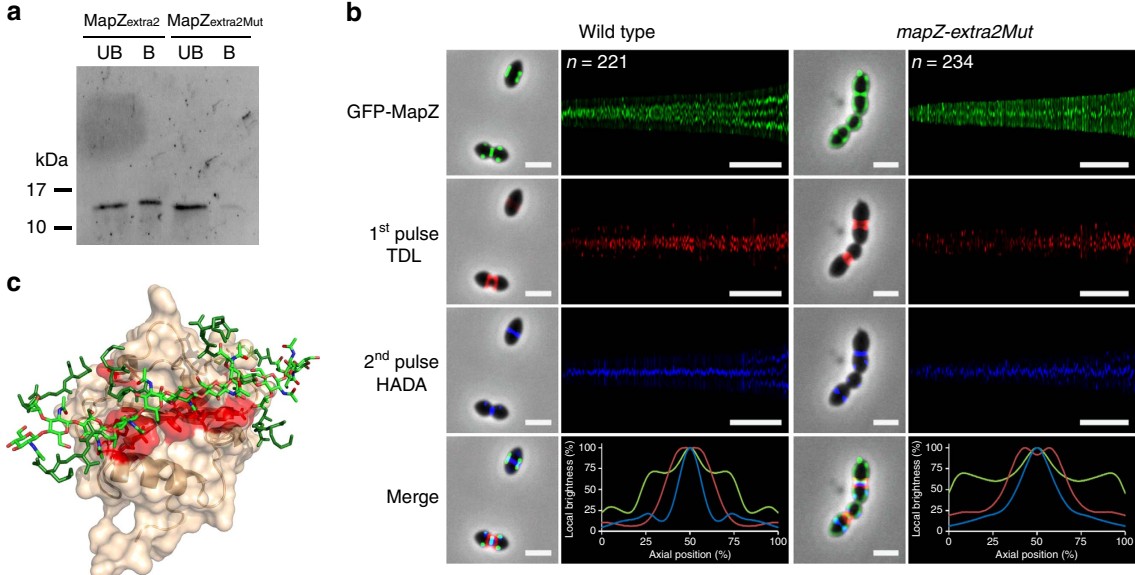

**Figure 6 | Impact of mutated MapZ$_{extra2}$ on peptidoglycan-mediated localization of MapZ.** (**a**) Interaction of MapZ$_{extra2}$ and MapZ$_{extra2-Mut}$ with the pneumococcal cell wall. The fraction of MapZ$_{extra2}$ or MapZ$_{extra2-Mut}$ unbound (UB) to cell wall and bound to cell wall was detected using a mouse anti-histidine-tag antibody. The experiment was made in triplicate. (**b**) Localization of peptidoglycan synthesis and MapZ in *gfp-mapZ* and *gfp-mapZ-extra2Mut* strains. Images of GFP fluorescence and peptidoglycan synthesis by two consecutive pulse-chase labellings, using the red fluorescent derivative of D-alanine TDL and then the blue fluorescent derivative of D-alanine HADA, are shown. An overlay of the three fluorescent labellings is shown at the bottom of the figure. Scale bars, 2 μm. The maps of fluorescence profiles for GFP-MapZ, HADA and TDL is also shown as in Fig. 5. Scale bars, 3 μm. The diagrams at the bottom of these columns show the relative distribution of fluorescence intensities of GFP-MapZ (green), TDL (red) and HADA (blue) along the cell length normalized to 100%. *n* indicates the number of cells analysed. (**c**) Lowest-energy structure obtained from the docking of 10 different peptidoglycan hexamuropeptide structures onto the MapZ$_{extra2}$ lowest-energy structure as described in Methods. All the residues mutated in the MapZ$_{extra2Mut}$ were considered as active Ambiguous Interaction Restraints during the HADDOCK minimization protocol and are coloured in red on the protein surface. Peptide stems and the glycosidic chain of the peptidoglycan are coloured in dark green and light green, respectively.

analysing the localization of GFP-MapZ devoid of MapZ$_{extra1}$ and SRL (strain *gfp-mapZΔextra1-ΔSRL*), it came thus as no surprise that MapZΔextra1-ΔSRL delocalized in the membrane and was not present at the cell equator (Fig. 5). Nevertheless, MapZΔextra1-ΔSRL positioned properly at the constricting septum. This localization pattern is similar to that observed in *gfp-mapZΔextra2* cells. As control, we checked that GFP-MapZΔextra1-ΔSRL was expressed at a physiological level (Supplementary Fig. 9).

Rather than reflecting an authentic role of MapZ$_{extra1}$ and the SRL in MapZ positioning at mid-cell, we reasoned that the deletion of *mapZ$_{extra1}$* and the *SRL* might generate a detrimental impact on MapZ$_{extra2}$ function. Indeed, when directly fused with the transmembrane span of MapZ, the position of MapZ$_{extra2}$ regarding the membrane and the peptidoglycan synthesis machinery is compromised. To circumvent this effect, and determine whether MapZ$_{extra1}$ is required *per se*, we substituted MapZ$_{extra1}$ for the N-terminal domain of the PBP1A activator LpoA (LpoA$_N$) of *E. coli*[21] (Fig. 4a). Like MapZ$_{extra1}$, the LpoA$_N$ domain possesses a short flexible N-terminal peptide that is followed by a compact α-helical domain, a tetratricopeptide repeat (TPR), of comparable stability and length to MapZ$_{extra1}$. Importantly, the amino-acid sequence of TPR domains is poorly conserved preventing very specific protein–protein interaction properties and excluding thus potential interference with the function of pneumococcal penicillin binding proteins (PBPs)[22]. The strain *mapZ-lpoA$_N$-extra2* thus expressed a MapZ-like protein, in which two flexible regions (the N-terminal end of the TPR and the SRL of MapZ) flanked a stable and anisotropic α-helical domain (the rigid α-helix bundle of the TPR). *mapZ-lpoA$_N$-extra2* cells were essentially indistinguishable from WT

cells, and displayed normal cell growth and viability (Fig. 4b; Supplementary Table 1; Supplementary Fig. 8), suggesting that MapZ-LpoA$_N$-extra2 is nearly fully functional for the cell division process. Indeed, all cells were rigorously shaped as WT cells. This shows that the insertion of LpoA$_N$ that includes a TPR motif largely suppresses the absence of MapZ$_{extra1}$ and restores the function of the MapZ$_{extra2}$ subdomain. Consistently, GFP-MapZ-LpoA$_N$-extra2 was properly placed at the division septum and the cell equator, and was produced normally (Fig. 5; Supplementary Fig. 9). Altogether, these data show that MapZ$_{extra1}$ can be substituted for the LpoA TPR motif, suggesting that the rigid MapZ$_{extra1}$ domain does not fulfill an essential function other than being a stable pedestal assisting the positioning MapZ$_{extra2}$ at the right place at the cell surface.

## Discussion

A couple of studies have now reported that different positive regulatory processes of cell division occur in bacteria[7,8,10]. Crucial questions remaining aim at determining how the cell centre is identified and reached by the components being part of these positive regulatory processes. Answering these questions has gained momentum with the characterization of the MapZ system in *S. pneumoniae*[10]. The pneumococcus divides using a semi-conservative cell growth process in which new peptidoglycan inserted at mid-cell forms the new cell halves of the two daughters cells, while the two old cell halves are moved apart. The membrane protein MapZ interacts with and is shuttled by the nascent peptidoglycan, and eventually localizes at the interface between the new and the old cell halves of the daughter cells. MapZ thus acts as a permanent molecular beacon of the

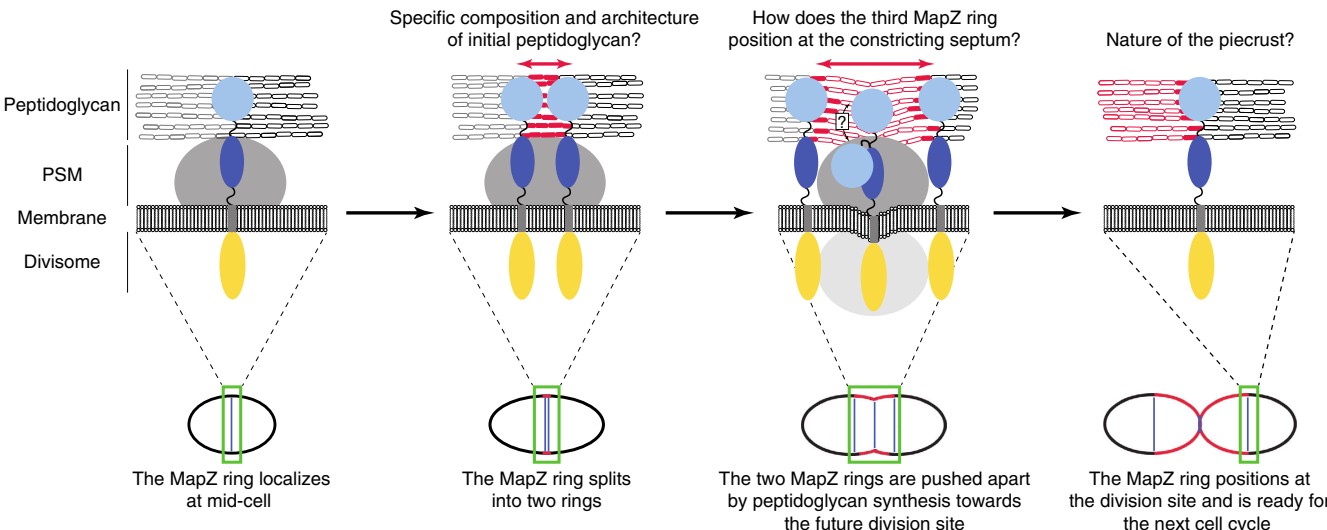

**Figure 7 | Model of MapZ positioning at the division site.** The upper part of the figure shows magnifications of the different cell division stages of the pneumococcus that are presented in the lower part. MapZ localizes at the interface between the two cell halves of newborn cells. Black and grey ovals distinguish the cell wall of each half-cell. The cytoplasmic domain of MapZ, MapZ$_{extra1}$ and MapZ$_{extra2}$ are represented by yellow, dark and light blue ovals, respectively. Initial peptidoglycan synthesis (red ovals) is recognized by MapZ$_{extra2}$ that is shifted towards the cell equator of the two daughter cells. Insertion of peptidoglycan allowing cell elongation and whose composition might be different from that of initial peptidoglycan is shown as open red ovals. The cytoplasmic part of the divisome and the peptidoglycan synthesis machinery (PSM) are shown in light and dark grey, respectively. For visual clarity, the divisome is not indicated in the first two stages of the cell division process. The question mark indicates the possible structural rearrangement of MapZ$_{extra}$ when localizing as a third ring at the constricting septum.

division site. In this study, we have solved the structure of the extracellular domain of MapZ and designed structure-based *in vivo* experiments to understand this process at the molecular level.

The MapZ extracellular domain is formed by two structured subdomains, with the N-terminal (MapZ$_{extra1}$) and C-terminal (MapZ$_{extra2}$) subdomains adopting an elongated shape of 55 Å in length and a spherical shape of ~30 Å in diameter, respectively. This organization as well as the fold of each domain shares no relevant structural similarity with any other proteins deposited in the PDB. The two structured domains are separated and connected by a flexible and disordered 42-amino-acid-long SRL. As shown by NMR and SPR, the two subdomains do not interact and behave as two independent modules with different rotational diffusion properties (Supplementary Figs 4 and 5). Such flexible SRLs are notably found in proteins involved in complex carbohydrate degradation in which it separates the catalytic domain and the carbohydrate-binding domain[23]. It is proposed that the SRL would enhance the substrate accessibility. This supports the view that MapZ$_{extra1}$ and MapZ$_{extra2}$ fulfill distinct functions and that the SRL serves to optimize the ability of MapZ$_{extra2}$ to bind peptidoglycan. In addition, the SAXS data fit with a $D_{max}$ of ~165 Å for the full-length MapZ$_{extra}$, suggesting that the two subdomains can extend in an overall rather elongated shape (Supplementary Fig. 5). Intrinsic flexibility of the SRL and the N-terminal end of MapZ$_{extra1}$ nevertheless prevented us to define a unique shape for the object and it can be proposed that MapZ could adopt different organization in the cell wall. As the exact distance between the membrane and MapZ$_{extra2}$ does not seem to be determinant for its function, it is thus tempting to speculate that this organization could confer the flexibility required for the spatial positioning of the C-terminal domain MapZ$_{extra2}$ for the subsequent interaction with peptidoglycan in the cell wall.

Interestingly, only MapZ$_{extra2}$ possesses a conserved and surface-exposed patch of hydrophilic amino acids, whereas the conserved amino acids in MapZ$_{extra1}$ are buried in the structure.

By combining this structural analysis and the *in vivo* behaviour of a mutant in which MapZ$_{extra1}$ is substituted for the N-terminal domain of LpoA on bacterial cell morphogenesis, we observe that the N-terminal domain of MapZ$_{extra}$ behaves mostly as a pedestal indispensable for the function of the C-terminal MapZ$_{extra2}$ subdomain (Figs 4 and 5). Nevertheless, this raises the question about the raison d'être for the stable and structured fold inside MapZ$_{extra1}$. It would be interesting to investigate the impact of MapZ$_{extra1}$ substitution in cell division, but also in other processes such as DNA uptake or polysaccharide capsule production, of less domesticated strains such as strains D39 or TIGR4 grown in different conditions. Indeed, capsule biosynthesis and DNA transformation occur at mid-cell and are coordinated with the cell cycle[24,25].

By contrast, mutations in the conserved and surface-exposed patch of MapZ$_{extra2}$ yield to a deficient interaction with peptidoglycan (Fig. 6a) as well as disruption of the proper localization of MapZ at the cell equator (Figs 5 and 6b). Docking of hexameric muropeptide structures, modelling the peptidoglycan, onto the MapZ$_{extra2}$ subdomain further supports the presence of a hydrogen-bond network between the saccharidic moieties of the peptidoglycan and the hydrophilic side chains of some of the mutated residues in the lowest-energy model (Fig. 6c). However, MapZ$_{extra2}$ does not interact with peptidoglycan from *E. coli* and *B. subtilis* (Supplementary Fig. 12), indicating that the mode of interaction between MapZ$_{extra2}$ and pneumococcal peptidoglycan requires specific features in the architecture and/or nature of the pneumococcal peptidoglycan. Future challenging solid-state NMR experiments should allow characterizing the peptidoglycan moieties in interaction with MapZ$_{extra2}$ and full understanding on this domain insertion in the cell wall[26]. Altogether, these results show that MapZ$_{extra2}$ is the key domain of MapZ to allow a stable and direct interaction between MapZ and the peptidoglycan polymer, and then MapZ shuttling from the division site to the cell equator of the daughter cell. This conclusion is corroborated by *in vivo* experiments showing that peptidoglycan synthesis at

mid-cell is not flanked by two outer-ring of MapZ$_{extra2Mut}$ (Fig. 6b). MapZ$_{extra2}$ could be considered as a novel protein fold binding peptidoglycan, thus increasing the existing repertoire of bacterial peptidoglycan binding domains[27–30]. An interesting observation is nevertheless that some of these domains are also positioned in the C-terminal end. This is notably the case of the SPOR domain of the cell division proteins FtsN, RlpA, DedD and DamX of *E. coli* that binds the denuded glycan chain of nascent peptidoglycan[31] or still the C-terminal PASTA domain of StkP of *S. pneumoniae* that is the only one to bind peptidoglycan sacculi[32].

One should however note that the deletion of *mapZ$_{extra2}$* as well as the mutation of the conserved patch of amino acids in MapZ$_{extra2}$ seems not to affect the ability of MapZ to position as a third ring at the current division site (Figs 5 and 6b). This suggests that the binding to peptidoglycan is not required for the localization of MapZ at the constricting septum. Instead, MapZ cytoplasmic subdomain could interact with cytoplasmic components of the divisome. As the cytoplasmic domain of MapZ is phosphorylated, the interaction with StkP represents a promising possibility. Localization of MapZ at the constricting septum also questions about the mode of insertion of the extracellular domain of MapZ across nascent peptidoglycan. MapZ localization as a third ring occurs, as cell elongation is not completed[10,33]. This implies that MapZ extracellular domain does not bind nascent peptidoglycan so as to maintain the protein at the constricting septum. We can speculate that the extracellular domain of MapZ could undergo structural rearrangement preventing binding of MapZ$_{extra2}$ to peptidoglycan. Alternatively, but not exclusively, the nature of the peptidoglycan produced at this stage of the cell cycle could differ from that of the peptidoglycan synthesized in newborn cells during initial cell elongation preventing MapZ$_{extra2}$ binding. This latter hypothesis is supported by observation of a 'piecrust' at the equatorial ring of pneumococcal cells. This piecrust is described as thicker bands of peptidoglycan localized at mid-cell at the site of the presumptive future septum[34] and could correspond to a heterogenic architecture and/or composition of peptidoglycan[35,36]. In addition, it is also documented that the composition of neosynthesized peptidoglycan differs from that of peptidoglycan sacculus[37].

Altogether, this study demonstrates that MapZ$_{extra1}$ serves as a pedestal to position properly MapZ$_{extra2}$ to bind peptidoglycan and to eventually localize at the equator of the daughter cell on completion of cell elongation. The model presented in Fig. 7 further suggests that MapZ$_{extra2}$ binding to initial peptidoglycan could be mediated by the particular composition and/or the architecture of the peptidoglycan forming the piecrust at the cell equator. Our data also imply that the third ring of MapZ would localize at the constricting septum on interaction with the divisome, while either structural rearrangements of the extracellular domain of MapZ or an alternative composition of peptidoglycan would contribute to stabilization at the constriction septum. Future work should allow testing this model and answering the questions raised to better understand the MapZ system.

## Methods

**Bacterial strains and growth conditions.** *S. pneumoniae* strains were cultivated at 37 °C in Todd Hewitt Yeast broth (Difco). Standard procedures for chromosomal transformation and viability assays were used[10,12,14,38]. Strains and plasmids, and primers used in this study are listed in Supplementary Tables 2 and 3, respectively.

**Allelic replacement mutagenesis and plasmid construction.** *Plasmids.* DNA fragments coding for MapZ$_{extra1}$ and MapZ$_{extra2}$ were obtained by PCR using chromosomal DNA from *S. pneumoniae* R800 as template. For MapZ$_{extra2Mut}$, gene amplification was performed using the chromosomal DNA of the strain *mapZ-extra2Mut*. Primers used are described in Supplementary Table 3. The obtained DNA fragments were cloned between the NdeI and BamHI cloning sites of the pETPhos plasmid[39]. The nucleotide sequences of all DNA fragments were checked to ensure error-free amplification.

*Mutant strains.* Throughout this study, gene mutagenesis, deletion or fusion with the *gfp* gene was constructed at each native chromosomal locus, expressed under the control of the native promoter, and represented the only source of protein in *S. pneumoniae*. All strains and primers are indicated in Supplementary Tables 2 and 3, respectively. The nucleotide sequences of all final PCR DNA fragments were checked to ensure error-free amplification.

To construct pneumococcus mutants (gene deletions, gene mutations or *gfp*-fusions), we used a two-step procedure, based on a bicistronic *kan-rpsL* cassette called Janus[38]. This procedure allows the replacement of a gene by a cassette and subsequent deletion or substitution of the cassette by a mutated allelic form at the gene chromosomal locus. Briefly, the Janus cassette is either used to replace the gene of interest or inserted at either its 5′ or 3′-end. Both options confer resistance to kanamycin and dominant streptomycin sensitivity in the WT streptomycin-resistant R800 *rpsL1* strain (Kan$^R$–Str$^S$). Then, any DNA fragment flanked on each end by sequences homologous to the upstream and downstream regions of the gene of interest are used to transform Kan$^R$–Str$^S$ strains to obtain the expected nonpolar markerless mutant strains.

In the following description, 'upstream' stands for the upstream region of *mapZ*, 'cyto' stands for the region coding for the cytoplasmic domain of MapZ, 'TM' stands for the region coding for the transmembrane domain of MapZ, 'extra1' stands for the region coding for MapZ$_{extra1}$, 'SRL' stands for the region coding for the SRL of MapZ, 'extra2' stands for the region coding for MapZ$_{extra2}$ and 'downstream' stands for the downstream region of *mapZ*.

To construct the *mapZΔextra1-ΔSRL* strain, the upstream + cyto + TM and the extra2 + downstream regions were amplified using the R800 strain as a template and the primer pairs 1/5 and 6/2, respectively. The obtained DNA fragments were used together in another fusion PCR using the primer pair 1/2. The resulting PCR product was used to transform the strain previously obtained[10] in which *mapZ* is replaced by the *kan-rpsL* cassette (R800-*mapZ::kan-rpsL*). This latter strain was also used as a recipient to generate all the *mapZ* mutants, fused or not to the *gfp*, in this article. The same procedure was applied to the strain *mapZΔextra2* with the primer pairs 1/7 (upstream + cyto + TM + extra1 + SRL) and 8/2 (downstream).

To construct the strain in which the MapZ$_{extra1}$ domain is replaced by LpoA$_N$ of *E. coli* (*mapZ-lpoA$_N$-extra2*), we first amplified the region coding for the LpoA$_N$ domain with the primer 9/10 and the *E. coli* K12 (XL1-Blue) strain as a template. Then, the upstream + cyto + TM and the SRL + extra2 + downstream regions were amplified using the R800 strain as a template and the primer pairs 1/5 and 11/2, respectively. The resulting DNA fragments 1–5 and 9–10 were used together in another fusion PCR using the primer pair 1/10. Finally, the resulting DNA fragments 1-10 and 11-2 were fused by PCR using the primer pair 1/2.

To construct the strain *mapZ-extra2Mut*, we first used the R800 strain chromosome as template and the primer pair 1/12 to introduce the mutations R409A-Y411A. The resulting DNA fragment 1-12 was then used as a primer with the primer 2 and the R800 strain as a template. Next, the resulting DNA fragment containing the mutations R409A-Y411A was then used as a template with the primer pair 1/13 to introduce the mutations N428A-Y430F. The resulting DNA fragment 1-13 was then used as a primer with the primer 2 and the R800 strain as a template. Last, the resulting DNA fragment containing the mutations R409A-Y411A-N428A-Y430F was used as a template with the primer 1/14 to introduce the mutations Y450A-F451L-N454A. The resulting DNA fragment 1-14 was then used as a primer with the primer 2 and the R800 strain as a template. Finally, we obtained a DNA fragment containing the whole locus of *mapZ*, with the mutations (R409A-Y411A-N428A-Y430F-Y450A-F451L-N454A).

To construct the two strains *ftsZ-gfp_mapZΔextra2* and *ftsZ-gfp_mapZ-extra2Mut*, the strain *ftsZ-gfp*[14] has been first transformed with a DNA fragment containing the locus of *mapZ* in which *mapZ* was replaced by the *kan-rpsL* cassette. Then, the strains *mapZΔextra2* and *mapZ-extra2Mut* were used as templates with the primer pair 1/2 to amplify the mutated *mapZ* loci. The resulting DNA products were used to transform the *ftsZ-gfp* Kan$^R$–Str$^S$ strain previously obtained in which *mapZ* was replaced by the *kan-rpsL* cassette.

For *gfp*-fusions, the upstream region of *mapZ* and the *gfp* gene were amplified using the primer pair 1/3 and the *gfp-mapZ* strain[10] as a template. The different mutated alleles of *mapZ* and the downstream region of *mapZ* were then amplified using the primer pair 4/2 and the strains previously obtained as templates. Then, the resulting DNA fragments 1-3 and 4-2 were fused by PCRs using the primer pair 1/2 and transformed in the pneumococcus. This procedure was applied for all *gfp* fusion strains.

**Protein production and purification.** Recombinant plasmids overproducing MapZ$_{extra}$-His$_6$, His$_7$-MapZ$_{extra1}$, His$_7$-MapZ$_{extra2}$ and His$_7$-MapZ$_{extra2Mut}$ were transformed into BL21(DE3) *E. coli* strain. The transformants were grown at 37 °C until the culture reached an $OD_{600} = 0.6$ in M9 medium supplemented with glucose and $NH_4Cl$ (ref. 21). After a first purification step performed with a Ni-NTA column (Qiagen)[10], the poly-His tag was cleaved by incubating the fusion protein with a His$_6$-tagged tobacco etch virus protease in a 40:1 (w/w) ratio in the presence of 0.5 mM EDTA and 1 mM dithiothreitol. The mixture was dialysed overnight at 4 °C, against a 25 mM Tris-HCl buffer at pH 7.5 containing 300 mM

NaCl, 10 mM imidazole, 5 mM β-mercaptoethanol and 10% glycerol. Cleaved His$_7$-MapZ$_{extra1}$, His$_7$-MapZ$_{extra2}$ and His$_7$-MapZ$_{extra2Mut}$ were then separated from the uncleaved fraction and tobacco etch virus protease by reloading the dialysed protein onto a Ni-NTA column. An additional gel filtration step was performed using a S75 16/600 column (GE Healthcare)[10]. The fractions corresponding to the pure protein were pooled and concentrated. The protein concentrations were determined using a Coomassie Assay Protein Dosage Reagent (Uptima) and aliquots were stored at $-80\,°C$. For NMR analysis, the C-terminal Histidine-tag in MapZ$_{extra}$ and the N-terminal Histidine-tag in each of the MapZ$_{extra1}$ and MapZ$_{extra2}$ subdomains were not cleaved for yield purposes on the isotopically labelled samples. Proteins for NMR studies were produced in minimal media containing isotopically labelled carbon ($^{13}$C-glucose) and nitrogen ($^{15}$NH$_4$Cl) sources[21].

**Immunoblot analysis.** Detection of GFP fusions[14] was performed using a rabbit anti-GFP polyclonal antibody (catalogue number TP401, AMS Biotechnology) at 1/10,000. Detection of the enolase[12] was performed using a rabbit anti-enolase polyclonal antibody (αEno, doi:10.1111/j.1365-2958.2011.07962.x) at 1/50,000. For cell wall binding assays, detection of MapZ$_{extra2}$ or MapZ$_{extra2Mut}$ was performed using a mouse monoclonal anti-6his antibody (catalogue number H1029, Sigma)[10]. A goat anti-rabbit secondary polyclonal antibody horseradish peroxidase conjugate (catalogue number 170–6515, Bio-Rad) was used at 1/5,000 to reveal immunoblots, except for the cell wall binding assay, in which the goat anti-mouse secondary antibody horseradish peroxidase conjugate (catalogue number 170–6516, Bio-Rad) was used at 1/5,000.

**Microscopy techniques.** Microscopy was performed on exponentially growing cells (OD$_{550}$ = 0.15). Slides were visualized using a Zeiss AxioObserver Z1 microscope fitted with an Orca-R2 C10600 charge-coupled device (CCD) camera (Hamamatsu), with a ×100 numerical aperture 1.46 objective. Images were collected using Axiovision (Carl Zeiss), convolved using ImageJ (http://rsb.info.nih.gov/ij/) and analysed using Coli-Inspector[40] (detection was approved manually) running under the plugin ObjectJ (http://simon.bio.uva.nl/objectj/) to generate fluorescent intensity linescans sorted with respect to cell length. The MicrobeTracker suite[41] extended by custom MATLAB routines was used to generate cell width/length ratio. Selected images are representative of experiments made in triplicate.

**NMR spectroscopy.** *Data collection.* NMR data were collected in 50 mM Tris, 100 mM NaCl, 10% D$_2$O, pH 7.5, at 298 K on 1.05, 2.0 and 2.5 mM $^{13}$C,$^{15}$N-MapZ$_{extra}$, MapZ$_{extra1}$ and MapZ$_{extra2}$ samples, respectively. All NMR spectra for backbone, side chains and nuclear Overhauser effect (NOE) assignments of MapZ$_{extra1}$ and MapZ$_{extra2}$ were recorded on Bruker spectrometers operating at 600, 700 and 950 MHz $^1$H NMR frequencies, and equipped with $^1$H,$^{13}$C,$^{15}$N-cryoprobes.

*Resonance assignments.* For both subdomains, manual assignments of the backbone were performed using 2D [$^1$H,$^{15}$N]-BEST-TROSY (BT), 3D BT-HNCO, 3D BT-HNCO+ (ref. 42), 3D BT-HNCACB and 3D BT-HN(CO)CACB spectra. Manual side-chain assignment was then achieved with conventional 2D [$^1$H,$^{13}$C]-HSQC, 3D (H)C(CCO)NH, 3D H(CCCO)NH, 3D $^{15}$N-NOESY-HSQC, as well as 3D aliphatic and aromatic $^{13}$C-NOESY-HSQC experiments (with a 140-ms mixing time for the three 3D NOESY-HSQCs). Spectra were processed with NMRPipe[43] and analysed with CcpNmr Analysis 2.4.1 (ref. 44).

*Extraction of structural restraints and structure calculation.* Dihedral angles (phi and psi) were predicted from backbone chemical shift by TALOS+ (ref. 45) and distance constraints were determined after manual peak-picking and automatic assignment of the 3D NOESY-HSQC experiments reported above by Unio10′ version 2.0.2 (ref. 46). Structures were subsequently calculated from these restraints by Aria 2.3.1 (ref. 18), with 100 structures from run 0–7, and 750 for the last one. The 20 lowest-energy structures were further refined in water and deposited in the PDB with accession numbers 2ND9 and 2NDA for MapZ$_{extra1}$ and MapZ$_{extra2}$ subdomains, respectively. Ramachandran analysis showed 90.0%, 9.9%, 0.1% and 0.0% of the residues of MapZ$_{extra1}$ in most favoured, additional allowed, generously allowed and disallowed regions, respectively. A similar analysis in MapZ$_{extra2}$ led to 84.4%, 14.3%, 0.1% and 1.2% of the residues of MapZ$_{extra2}$ in most favoured, additional allowed, generously allowed and disallowed regions, respectively.

*NMR interaction assay between MapZ$_{extra1}$ and MapZ$_{extra2}$.* To evaluate the interaction between MapZ$_{extra1}$ and MapZ$_{extra2}$ subdomains, 2D [$^1$H,$^{15}$N]-BEST-TROSY spectra were collected on a 100 μM sample of $^{13}$C,$^{15}$N-labelled MapZ$_{extra1}$ before and after addition of two molar equivalents of $^{13}$C,$^{15}$N-labelled MapZ$_{extra2}$. These NMR spectra were recorded in the same buffer and temperature conditions than the spectra collected for structure determination purposes on the individual subdomains.

*Relaxation measurements.* Relaxation studies were performed on the $^{13}$C,$^{15}$N-MapZ$_{extra1}$ and MapZ$_{extra2}$ samples used for structure determination, as well as on the full-length $^{13}$C,$^{15}$N-MapZ$_{extra}$ sample. Longitudinal $R_1$, transverse $R_2$ and {$^1$H}$^{15}$N-NOE relaxation data were collected at 25 °C on Bruker spectrometers operating at 600 MHz and equipped with $^1$H,$^{13}$C,$^{15}$N-cryoprobes. Relaxation delays of 0, 0.01, 0.025, 0.05, 0.1, 0.2, 0.4, 0.8, 1.0 s and 0, 0.017, 0.034, 0.068, 0.085, 0.102, 0.119, 0.136, 0.170, 0.204, 0.237 s were used to determine longitudinal and

transverse relaxation rate constants, respectively. For each amide resonance, $R_1$ and $R_2$ were fitted from the evolution of the resonance intensity as a function of the relaxation delay by the dedicated module in CcpNmr Analysis 2.4.1 (ref. 44). Standard deviations on these values were calculated from Monte Carlo simulations during the fitting procedures. {$^1$H}$^{15}$N-NOE values were determined by the comparison of the intensities of each amide resonance with and without a 3-s saturation period. Standard deviations were calculated from errors on peak intensities.

**Surface plasmon resonance analyses.** Real-time binding experiments[10] were performed on a BIAcoreT100 biosensor system (GE Healthcare). Briefly, MapZ$_{extra2}$ was covalently coupled with the surface of a CM5 sensorchip and increasing concentrations (0.01, 0.02, 0.05, 0.1, 0.2 and 0.5 μM from bottom to top) of MapZ$_{extra1}$ were injected over the surface of the sensorchip at a flow rate of 30 μl min$^{-1}$ in 10 mM HEPES (pH 7.4), 150 mM NaCl and 0.005% surfactant P20. Non-specific binding to the surface of the sensorchip was subtracted by injection of the analytes over a mocked derivatized sensorchip.

**Small-angle X-ray scattering.** SAXS data were collected on beamline BM29 from the European Synchrotron Radiation Facility in Grenoble (France) on two MapZ$_{extra}$ samples at concentrations of 5 and 2 mg ml$^{-1}$ in the same buffer as for the NMR experiments. Ten frames of 1 s each were recorded on each sample, positioned 2.86 m from a Pilatus detector, at a wavelength of 0.99 Å. For each sample, frames were normalized to the intensity of the transmitted beam before being merged. Buffer's contribution to the scattering was then subtracted using the PRIMUS software from the ATSAS 2.5.1 software package[47]. Radius of gyration, $R_g$, forward scattering intensity, $I(0)$, maximum particle dimension, $D_{max}$, and distance distribution function, $P(r)$, were evaluated with GNOM from the same program suite.

**Peptidoglycan labelling and binding assays.** The procedure used to label peptidoglycan with fluorescent D-amino acids (FDAAs) and perform virtual time-lapse microscopy was adapted from reference[20] and performed as described in reference[10]. TDL and HADA were the two FDAAs used in this study. Briefly, exponentially growing *gfp-mapZ* strains (OD$_{550}$ = 0.1; Supplementary Table 2) were incubated for 1 min at 37 °C in Todd Hewitt Yeast broth (Difco) with 500 μM of TDL (a fluorescent carboxytetramethylrhodamine derivative of D-alanine). Cells were then washed three times with 1 ml PBS at pH 7.4 and room temperature, concentrated to OD$_{550}$ = 0.1, incubated again for 1 min at 37 °C with 500 μM of HADA (a fluorescent hydroxy coumarin derivative of D-alanine) and washed three times with PBS. A measure of 0.7 μl of each mixture was then placed on slides and observed under the microscope. These experiments were made in triplicates. Time-lapse microscopy was performed as described[10] using an automated inverted epifluorescence microscope Nikon Ti-E/B equipped with the perfect focus system (Nikon) and a phase-contrast objective (CFI Plan Fluor DLL, ×100 oil numerical aperture 1.3), a Semrock filter set for GFP (Ex: 482BP35; DM: 506; Em: 536BP40), a Nikon Intensilight 130 W High-Pressure Mercury Lamp, a monochrome Orca-R2 digital CCD camera (Hamamatsu) and an ImagEM-1 K EMCCD camera (Hamamatsu). The microscope is equipped with a chamber thermostated at 30 °C. Images were captured every 5 min and processed using Nis-Elements AR software (Nikon). All fluorescence images were acquired with a minimal exposure time to minimize bleaching and phototoxicity effects. GFP fluorescence images were false coloured green and overlaid on phase-contrast images. Pneumococcal cell wall preparation was performed as described in references[10,32]. Briefly, a 2-l culture of *S. pneumoniae* R6 cells in Todd Hewitt medium (BD Sciences) was incubated at 30 °C until OD$_{600}$ = 0.5. Cells were collected by centrifugation for 10 min at 4 °C and 7,500g and were resuspended in 40 ml of ice-cold 50 mM Tris at pH 7.0. The cell suspension was poured dropwise into 150 ml of boiling 5% SDS solution and boiled for another 30 min. Cell debris were pelleted by centrifugation for 20 min at 20 °C and 48,384g. The pellet was washed twice with 25 ml of 1 M NaCl for 30 min at 20 °C and 48,384g, and repeatedly with water until it was free of SDS as checked by Hayashi test. The pellet was resuspended in 20 ml of buffer containing 100 mM Tris at pH 7.5, 20 mM MgSO$_4$, 10 μg ml$^{-1}$ DNase A and 50 μg ml$^{-1}$ RNase I, and incubated for 2 h at 37 °C with gentle shaking. After addition of 10 mM CaCl$_2$ (final concentration) and 100 μg ml$^{-1}$ trypsin, the incubation was pursued for 18 h at 37 °C with gentle shaking. Proteases were inactivated with addition of SDS (1% final concentration) and incubation for 15 min at 80 °C. The cell wall preparation was recovered by ultracentrifugation for 30 min at 25 °C and 257,320g, resuspended in 20 ml of 8 M LiCl, and incubated for 15 min at 37 °C. After another ultracentrifugation step, the pellet was resuspended in 25 ml of 10 mM EDTA pH 7.0 and incubated at 37 °C for 1 h. The cell wall was washed with water before being resuspended in 2 ml of water and conserved at 4 °C. Peptidoglycan sacculi from *E. coli* and *B. subtilis* were purchased from InvivoGen (catalogue reference tlrl-pgnek) and Sigma (catalogue number 69554), respectively.

Binding of the pneumococcal cell wall preparation to MapZ$_{extra2}$ and MapZ$_{extra2Mut}$ was performed according to the procedures described in references[10,32]. Briefly, purified MapZ subdomains (10 μg ml$^{-1}$) were incubated with purified cell wall (7 mg) in 50 μl of 50 mM Tris pH 8.0, 100 mM NaCl for 16 h at 4 °C. After centrifugation (5 min at 5,000g), the supernatant was removed and

the cell wall pellet was washed three times and resuspended in 50 μl Laemmli buffer. After an incubation at 100 °C for 10 min and a centrifugation step (3 min at 5,000g), the supernatant fraction, corresponding to the bound protein fraction, was recovered from the cell wall pellet. Unbound and bound fractions were analysed by SDS–polyacrylamide gel electrophoresis subjected to western blotting using mouse anti-histidine-tag antibody.

**Data-driven docking of MapZ$_{extra2}$ with muropeptides.** The peptidoglycan model is a sixfold repetition of an $N$-acetylglucosamine (GlcNAc)-$N$-acetylmuramic acid (MurNAc)-L-Ala$_1$-γ-D-iGln$_2$-L-Lys$_3$ S. pneumoniae muropeptide. Topology, parameter files and patches for CNS were generated using the GLYCANS software (http://haddock.chem.uu.nl/enmr/services/GLYCANS/) for the oligosaccharidic backbone and complemented to integrate the lactoyl-peptide stems. A starting hexamuropeptide structure was generated with CNS[17] and minimized in explicit water molecules, while restraining the φ and ψ dihedral angles of the β-1,4-glycosidic bond to 69° and 12° (±30°), respectively[48]. From this initial structure, an ensemble of ten low-energy structures was generated by simulated annealing[49]. A model of the muropeptide–MapZ$_{extra2}$ complex was then calculated using HADDOCK 2.1 (ref. 49) by docking the 10 obtained hexamuropeptide structures onto the structure of the MapZ$_{extra2}$ subdomain determined by NMR in this study. All of the atoms of the muropeptide hexamer were considered as passive Ambiguous Interaction Restraints (AIR), while residues mutated in the MapZ$_{extra2Mut}$ were considered as active AIRs. The calculation was run with 4,000 structures during the rigid body energy minimization, 600 structures during the refinement and 200 structures during the refinement in explicit water. The output peptidoglycan:MapZ model structures were sorted with the HADDOCK built-in clustering tool using a 10-Å cutoff and a minimum of three structures per cluster.

**Data availability.** NMR chemical shifts and resonance assignments, and atom coordinates are available at the Biological Magnetic Resonance Bank (www.bmrb.wisc.edu) and Protein Data Bank (www.rcsb.org), respectively, under the accession numbers 26052 and 2ND9 for MapZ$_{extra1}$ and 26053 and 2NDA for MapZ$_{extra2}$. All other data are available from the corresponding authors on reasonable request.

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

## Acknowledgements

This work was supported by grants from the Agence Nationale de la Recherche (ANR-12-BSV3-0008-01 and ANR-15-CE32-0001). N.L.J. was supported by a PhD fellowship from the CEA. We acknowledge the contribution of the PLATIM and Protein Sciences platforms of SFR Biosciences Gerland-Lyon Sud (UMS344/US8). This work used the NMR and isotope labelling platforms of the Grenoble Instruct Center (ISBG; UMS 3518 CNRS-CEA-UJF-EMBL) with support from FRISBI (ANR-10-INSB-05-02), GRAL (ANR-10-LABX-49-01) and a financial support from the TGIR-RMN-THC FR3050. Access to the ESRF SAXS beamline BM29 and technical support for data collection is acknowledged. We gratefully thank Yves V. Brun for stimulating discussions and help in the use of FDAAs.

## Author contributions

S.M. conducted all experiments of cell biology and genetics. N.L.J. conducted the NMR experiments, and M.G. expressed and purified all MapZ domains and performed western blot analysis. J-P.L. performed the SPR experiments. M.S.V. provided new reagents. S.M., N.L.J., M.G., J-P.L., C. L. and C.M.B. designed and analysed the data together with C.G. and J.-P.S. C.G. and J.-P.S. wrote the manuscript and all authors edited the manuscript.

## Additional information

**Competing financial interests:** The authors declare no competing financial interests.

