## [Peer review file · Nature Communications]

Reviewers' Comments:

Reviewer #1 (Remarks to the Author)

In this manuscript, the authors investigate the molecular characterization of the extra-cytoplasmic part of MapZ, a newly described positive regulatory protein of bacterial cell division. Notably, they reveal the structure of the extra-cytoplasmic part of MapZ and show that it is composed by two different subdomains (mapZextra1 and MapZextra2) separated by a flexible serine-rich linker. According to the structural data obtained, they dissect the function of these two domains in vivo. The deletion of the second domain (MapZextra2) recapitulates the phenotype of a mapZ mutant strain with a lot of minicell and aberrant cell division localization. An examination of the localization of this variants confirms (as already shown in the Nature paper) that the extra-cytoplasmic part of MapZ controls its localization, but now, the authors show that the MapZextra2 is essential for MapZ localization using a heptuple point mutant of several surface exposed amino acids (MapZ-extra2mut). This variant recapitulates the phenotype of MapZextra2 deletion mutant with also a defect in MapZ localization. In addition, these amino acids are important for the binding of the peptidoglycan. Finally and remarkably, the authors show that mapZextra1 and the flexible serine-rich linker are like a pedestal allowing the binding of peptidoglycan by mapZextra2. Nevertheless, the presence of a folded domain (mapZextra1) is not yet understood, will any PG binding domain work? If not what is the specificity of PG binding (see comment 4). Overall data are convincing and the experiments well executed, but the story could be explored further a bit for a Nature Communications story.

I do have some reservations about the interpretation of the HADA staining in the MapZextra2 mutant (Fig. 6B) as I do not find a striking difference between PG synthesis. Some of the precision may be lost, but I wonder if the MapZextra2 mutant is dominant negative that affects PG synthesis or turnover in other ways. Is the cytoplasmic domain required for this (possibly dominant negative) effect? I also find the repeated references to MapZ being "pushed", "shifting" or "moving" by PG an overinterpretation. If this is the case, why does it stop at the quarter positions? Does PG synthesis cease at this time? The statement that "MapZ is thus a dualfunction protein, which not only serves to mark the division site and position FtsZ, but that also controls the Z-ring closure" is also an overinterpretation leaning on negative results. At this point it cannot be excluded that these effects are consequences of the same activity (defect). As the authors state it would be also nice to explore the effects of the MapZextra2, the LpoA or a cytoplasmic deletion derivative for capsule synthesis and competence to rule out pleiotropic effects via these pathways....

Comments

- 1/ Figure4: Could the authors test the localization of FtsZ for the different variants?
- 2/ Figure5: It seems that two independent mechanisms allow MapZ localization, one dependent of the peptidoglycan and the other (at the constriction site) independent of the PG. Are both lost in the FtsZ depletion strain? Does the author have any explanation for this? What happens when the phosphoablative mutations are combined with MapZextra1 or MapZextra2 mutation?
- 3/ Is the MapZextra2 a dominant negative (inducing the cell division defects also in a mapZ+ context)?
- 4/ Is it known if the binding to peptidoglycan by MapZ is specific to *S. pneumoniae* peptidoglycan? Is it possible to test the interaction of MapZ with peptidoglycan from other bacteria (ideally from a gram positive bacterium and a gram negative bacterium)?

Minor comments

- 1/ page9- Table S2: could the authors show the result for a mapZ mutant strain?

2/ page10-Figure S8: with the representation shown, the difference between the WT and the different variants are not really convincing?

3/ Figure S9 does not show if GFP is clipped off as GFP (25 kDa) is cut off the blot!!!!

Reviewer #2 (Remarks to the Author)

The present work by Manuse et al. provides the NMR structure of the extracellular domain of MapZ and in addition a comprehensive analysis of the function of this domain based on structure-inspired *in vivo* experiments. It is convincingly shown that the extracellular domain of MapZ consists of two independent subdomains connected by a flexible serine-rich linker and that the N-terminal subdomain functions as a pedestal for the C-terminal subdomain which mediates the interaction with peptidoglycan. I suggest acceptance of this nice work in Nature Communications provided that the following issues can be addressed:

- Introduction: "These two new systems..." Provide more details. What are these two new systems?
- Results: "The characteristic signatures..." Show in the main text a figure that contains an overlay of the spectra (or excerpts of them) of the individual subdomains and the complete domain (with different colors) in order to demonstrate that there are no changes.
- R2/R1 values (p.7): Provide Errors (SD). Did the authors measure R2/R1 also for the individual subdomains? How do these values compare to the full-length construct?
- p.13: "Unexpectedly and strikingly..." Why is this unexpected? Wasn't the experiment designed to produce this effect?
- Maybe for the discussion: Can the serines in the linker be phosphorylated and if yes what would be the expected result?

Reviewer #1:

In this manuscript, the authors investigate the molecular characterization of the extra-cytoplasmic part of MapZ, a newly described positive regulatory protein of bacterial cell division. Notably, they reveal the structure of the extra-cytoplasmic part of MapZ and show that it is composed by two different subdomains (MapZ_{extra1} and MapZ_{extra2}) separated by a flexible serine-rich linker. According to the structural data obtained, they dissect the function of these two domains *in vivo*. The deletion of the second domain (MapZ_{extra2}) recapitulates the phenotype of a *mapZ* mutant strain with a lot of minicell and aberrant cell division localization. An examination of the localization of this variants confirms (as already shown in the Nature paper) that the extra-cytoplasmic part of MapZ controls its localization, but now, the authors show that the MapZ_{extra2} is essential for MapZ localization using a heptuple point mutant of several surface exposed amino acids (MapZ_{extra2mut}). This variant recapitulates the phenotype of MapZ_{extra2} deletion mutant with also a defect in MapZ localization. In addition, these amino acids are important for the binding of the peptidoglycan. Finally and remarkably, the authors show that MapZ_{extra1} and the flexible serine-rich linker are like a pedestal allowing the binding of peptidoglycan by MapZ_{extra2}. Nevertheless, the presence of a folded domain (MapZ_{extra1}) is not yet understood, will any PG binding domain work? If not what is the specificity of PG binding (see comment 4). Overall data are convincing and the experiments well executed, but the story could be explored further a bit for a Nature Communications story.

We agree with the referee that the presence of the folded domain MapZ_{extra1} is not yet completely understood, as it can be replaced by the folded domain of the TPR domain of LpoA. To check whether it can be replaced by any PG binding domains, we have considered different possibilities and finally opted for the MapZ_{extra2} domain as it is now one of the best characterized in the pneumococcus. We have thus constructed a strain expressing MapZ with two MapZ_{extra2} domains in tandem and separated by the SRL (MapZ_{extra2-extra2}). The same strain was constructed with this form of MapZ fused to the GFP (GFP-MapZ_{extra2-extra2}). Both types of strains grow properly and display normal cell shape. In addition, the localization pattern of GFP-MapZ_{extra2-extra2} is similar to that of GFP-MapZ in WT cells. These data are shown below but not in the manuscript because we do not think they contribute to a better understanding of the role of MapZ_{extra1}. Rather, and as hypothesized in the discussion part of the manuscript, we suggest that, beyond the function of MapZ_{extra1} as a pedestal for MapZ_{extra2}, MapZ_{extra1} might be required for other processes connected to cell division in less domesticated strains. We agree that future studies should address these questions. Please read also our answer to the referee comment 4 below.

I do have some reservations about the interpretation of the HADA staining in the MapZ_{extra2} mutant (Fig. 6B) as I do not find a striking difference between PG synthesis. Some of the precision may be lost, but I wonder if the MapZ_{extra2} mutant is dominant negative that affects PG synthesis or turnover in other ways. Is the cytoplasmic domain required for this (possibly dominant negative) effect?

PG synthesis occurs at mid-cell and does not require MapZ (Fleurie et al, Nature, 2014). In that context, the absence of either MapZ_{extra2} or its cytoplasmic domain do not have any impact on PG synthesis that is coordinated with and organized by FtsZ (Fleurie et al, PLoS Genet. 2014). Therefore, and as stated by the referee, it came as no surprise that the pattern of PG synthesis tracked with HADA and TDL is similar in WT and *mapZ-extra2Mut* cells (Figure 6B). Rather and importantly, Figure 6B shows that proper PG synthesis in *mapZ-extra2Mut* cells is unable to "shuttle" MapZ toward the cell equators (future division site of daughter cells at position 25 and 75 (1/4;3/4) in the WT strain). Indeed, the MapZ fluorescence signal is distributed all along *mapZ-extra2Mut* cells and not confined at position 25 and 75 anymore.

I also find the repeated references to MapZ being "pushed", "shifting" or "moving" by PG an overinterpretation. If this is the case, why does it stop at the quarter positions? Does PG synthesis cease at this time?

The data presented in this manuscript together with that published in Fleurie et al., Nature 2014 and Holeckova et al., Mbio, 2015, supports our conclusion. Once MapZ is bound to peptidoglycan, the continuous production of PG at mid-cell, elongating the cell and forming the new cell halves of the two daughter cells, "pushes" MapZ that eventually positioned at the cell equator when cell elongation stops. PG synthesis actually never stops but serves either cell elongation or cell division (synthesis of the cross-wall) when cell elongation is completed. In other words, MapZ permanent association with the cell equator is entangled with the fact that *Streptococcus pneumoniae* cells elongate from mid-cell, resulting in what could be termed a "semi-conservative cell growth". PG-synthesis builds the new cell-halves from the cell center while the intact old cell-halves are pushed apart. This process consequently moves away the MapZ rings that are located at the frontier between new and old cell wall material, *i.e.*, at the cell equator.

We still do not understand how cell elongation is regulated and coordinated with cell division. From our previous study (Fleurie et al., PLoS Genet., 2014) showing that the three cell division proteins DivIVA, GpsB and StkP act as a molecular switch coordinating cell elongation with cell constriction, an interesting hypothesis would be that StkP-mediated phosphorylation of some phosphorylation targets, including DivIVA, would act as a "clock" favoring first the production of PG for cell elongation until the appropriated cell size is reached and then the synthesis of the cross-wall to allow separation of the two daughter cells.

The statement that "MapZ is thus a dualfunction protein, which not only serves to mark the division site and position FtsZ, but that also controls the Z-ring closure" is also an overinterpretation leaning on negative results. At this point it cannot be excluded that these effects are consequences of the same activity (defect).

We disagree. Our previous observations (Fleurie et al., Nature 2014) show that MapZ is required for the proper positioning of FtsZ at mid-cell. In the same study, it is also shown that the constriction of the Z-ring is compromised in the absence of MapZ or when MapZ phosphorylation is deregulated. Indeed, MapZ is able to fulfill its first function (positioning of the divisome at mid-cell) but not the second function (proper constriction of the Z-ring) in phospho-ablative and -mimetic *mapZ* mutants.

As the authors state it would be also nice to explore the effects of the MapZ_{extra2}, the LpoA or a cytoplasmic deletion derivative for capsule synthesis and competence to rule out pleiotropic effects via these pathways....

We agree with the referee that exploring the effects of MapZ in other processes such as capsule synthesis or DNA uptake would be nice. Such a goal is however challenging and

represents tremendous pieces of work. For instance, our study was performed in the laboratory R6 strain that is not capsulated. In addition, cells were not grown in conditions in which competence is induced. Investigating the impact of MapZ on capsule production and DNA uptake would require to do again all the same analyses after having constructed again all *mapZ* mutants in capsulated strains and made them grow in conditions triggering competence. In addition, and importantly, a recent study has shown that the effect of *mapZ* deletion is hidden in capsulated strains (Boersma et al., J. Bact 2015). Moreover, induction of competence induces a transient growth arrest and stop in cell wall synthesis (Seto and Tomasz, J. Bact, 1964 and Mirouze et al., PNAS, 2004). In both cases, the underlying regulatory mechanisms remain enigmatic. Deciphering these mechanisms is thus a prerequisite before tackling the potential role of MapZ in coordinating cell division with other cellular processes.

Comments

1/ Figure4: Could the authors test the localization of FtsZ for the different variants?

Done. We now present as Supplementary Figure 10 (referenced on page 10) the localization of FtsZ-GFP (which is functional and expressed as the sole source of FtsZ in cells (Fleurie et al, PLoS Genet., 2014)) in WT, *mapZextra2* and *mapZextra2Mut* cells. In the two later strains, the localization of FtsZ is compromised showing that a functional MapZ_{extra2} domain is needed for the localization of FtsZ at mid-cell.

2/ Figure5: It seems that two independent mechanisms allow MapZ localization, one dependent of the peptidoglycan and the other (at the constriction site) independent of the PG. Are both lost in the FtsZ depletion strain?

This is an interesting question that we have tried to address. To test the effect of FtsZ depletion, we first inserted a copy of wild-type *ftsZ* at the ectopic *bgaA* locus under the control of the P_{Zn} promoter (P_{Zn}-*ftsZ*). Then we tried to knock-out the chromosomal copy of *ftsZ*. However, despite our efforts we never got transformants indicating that the P_{Zn} promoter does not allow appropriate level of FtsZ expression. To overcome this problem, we tried to use the pCEP platform (Guiral et al, microbiology, 2005), which integrates in the chromosome downstream of the *ami* operon and harbors the maltose-inducible P_M promoter. However, we encountered similar expression issues. We are thus unable to evaluate the impact of FtsZ depletion on MapZ localization. FtsZ being an essential gene, it is particularly difficult to delete even in model organism in which a variety of genetic tools are available. In our case, the two genetic systems we tried failed leading to satisfactory FtsZ depletion. One can however anticipate that pneumococcal cells expressing low level of FtsZ will be aberrantly shaped and that MapZ localization should be altered, as many other division proteins.

We agree with the referee that two independent mechanisms allow MapZ localization. In this manuscript, we think that we convincingly show that MapZ localization at the cell equator depends on its ability to bind peptidoglycan via the domain MapZ_{extra2}. By contrast, how MapZ positions as a third ring at the division site remains enigmatic. As stated by the referee, one can suggest that it would be independent of PG synthesis. However, the other way round cannot be ruled out as septal PG synthesis is required for cell constriction and formation of the cross-wall. In addition, the use of antibiotics such as vancomycin, that inhibit PG synthesis, induces rapid delocalization of MapZ (Fleurie et al., Nature, 2014). The data presented in this paper however suggests that the domains Extra2 and Extra1 are not essential for the formation of the third ring. Hence, further work is needed to determine the importance of the transmembrane spans as well as the cytoplasmic domain in MapZ localization at the constricting septum.

Does the author have any explanation for this?

Our data show that MapZ is required at two separate moments in time in the cell: i) to position the future division site and ii) to regulate cell constriction. On this basis, it is hardly conceivable that the same mechanism would be responsible for this dual localization. As demonstrated in this study, one mechanism is based on the interaction with peptidoglycan to allow MapZ beaconing of the division site. Further work is needed to decipher the second mechanism that is required to position MapZ at the septum to control cell constriction. From this point, many speculations can be made for the second mechanism. For instance, MapZ could actually interact with any components of the divisome, including StkP that phosphorylates MapZ. The situation can be even more complex and one cannot exclude that the second mechanism would also be based on peptidoglycan binding as well. Indeed, the nature of the peptidoglycan recognized by MapZ is still unknown. In this scheme, MapZ might recognize PG produced only at the beginning of cell elongation and at the beginning of cell constriction. An alternative might be that MapZ might recognize two distinct PG fragments produced at two different moments in the course of the cell cycle, either during cell elongation or when starting cell constriction. Complicated as it may, further work is needed to check these hypotheses.

What happens when the phosphoablative mutations are combined with MapZ_{extra1} or MapZ_{extra2} mutation?

We have constructed these strains but we were not able to observe further defects. Cells expressing MapZ devoid of the Extra1 or Extra2 domains are actually already strongly affected in cell shape, viability and generation time, and we were not able to detect further obvious detrimental effects in the presence of phospho-ablative or -mimetic mutations.

3/ Is the MapZ_{extra2} a dominant negative (inducing the cell division defects also in a mapZ⁺ context)?

We do not think that this experiment would provide further information on the role of the cytoplasmic or extracellular domains of MapZ in absence of a better biochemical characterization of the cytoplasmic domain of MapZ. In the worst-case, one can anticipate that the expression of a truncated version of MapZ will interfere with the WT copy of MapZ and would have a detrimental impact on the cell. Another important issue with such an experiment would be the global level of expression of MapZ that would be higher than in WT cells. On the other hand, the ratio MapZ/MapZ Δ extra2 would not be precisely controlled as well. In both cases, they could distort the interpretation of the observations.

4/ Is it known if the binding to peptidoglycan by MapZ is specific to *S. pneumoniae* peptidoglycan? Is it possible to test the interaction of MapZ with peptidoglycan from other bacteria (ideally from a gram positive bacterium and a gram negative bacterium)?

Done. We present in new Supplementary Figure 12 (referenced on page 11 and 12) that MapZ_{extra2} binding to PG from *E. coli* and *B. subtilis* is nearly nonexistent. The same experiment was performed with MapZ_{extra2Mut} and led to the same observation. Together with data presented in Figure 6A, this shows that MapZ_{extra2} recognized specifically the PG of the pneumococcus.

Minor comments

1/ page9- Table S2: could the authors show the result for a mapZ mutant strain?

This result is actually already presented in Fleurie et al., Nature, 2014. For an easy flow, we have added again the viability and the generation time of the *mapZ* mutant in this

Supplementary Table 2.

2/ page10-Figure S8: with the representation shown, the difference between the WT and the different variants are not really convincing?

We have now added another representation (histogram showing the frequency of the different length/width ratio) in Supplementary Figure 8. Together with the dot-cloud representation, we think that the difference between the WT and the different variants is clearly shown.

3/ Figure S9 does not show if GFP is clipped off as GFP (25 kDa) is cut off the blot!!!!

Done. We now show the lower part of the gel.

Reviewer #2:

The present work by Manuse et al. provides the NMR structure of the extracellular domain of MapZ and in addition a comprehensive analysis of the function of this domain based on structure-inspired in vivo experiments. It is convincingly shown that the extracellular domain of MapZ consists of two independent subdomains connected by a flexible serine-rich linker and that the N-terminal subdomain functions as a pedestal for the C-terminal subdomain which mediates the interaction with peptidoglycan. I suggest acceptance of this nice work in Nature Communications provided that the following issues can be addressed:

- Introduction: "These two new systems..." Provide more details. What are these two new systems?

We agree. We now cite on page 3 the two systems, PomZ and SsgA/SsgB, found in *M. xanthus* and *S. coelicolor*.

- Results: "The characteristic signatures..." Show in the main text a figure that contains an overlay of the spectra (or excerpts of them) of the individual subdomains and the complete domain (with different colors) in order to demonstrate that there are no changes.

We tried to superimpose the 2D NMR spectra of MapZ_{extra1}, MapZ_{extra2} and the full length MapZ constructs with 119, 97 and 258 peaks, respectively. Nevertheless, the figure becomes very unclear due to the total number of peaks to be displayed. We also had to take into account that some of the peaks show a slight shift due to (i) the different positions of the His-tag used for purification purposes in the various protein constructs and (ii) the presence of the serine-rich linker (SRL) that exclusively occurs in the full-length construct.

As a consequence, we decided to modify only slightly panel B of Figure 1 by including (i) the resonance assignment in the spectra of the individual subdomains MapZ_{extra1} and MapZ_{extra2} and (ii) a partial assignment of the resolved resonances in the spectrum of the full-length form with some horizontal lines showing the correspondence between the isolated subdomains and the full-length construct. However, we added a new panel C in Figure 1 with two different excerpts of the superimposition of MapZ_{extra1} (green), MapZ_{extra2} (red) and MapZ_{extra} (black) spectra, as suggested by the referee. This new panel evidences the overlay of resonances of the full-length construct with those of the individual domains, except for residue K308 which experiences a significant shift as expected from its position at the N-terminus of MapZ_{extra1} and the resonance of Q316 which comes from the SRL and has no equivalent in the individual subdomains (see excerpt on the right of panel C). These results show that the structure of the MapZ_{extra1} and MapZ_{extra2} subdomains does not change significantly in the full-length construct with respect to the isolated subdomains. The legend of this overall new Fig. 1 has

been updated accordingly.

- R₂/R₁ values (p.7): Provide Errors (SD). Did the authors measure R₂/R₁ also for the individual subdomains? How do these values compare to the full-length construct?

The R_2/R_1 values and their standard deviations, shown as vertical blue histograms and black bars, respectively, are provided for the individual MapZ_{extra1} and MapZ_{extra2} subdomains in Supplementary Figure 6 and for the full-length MapZ_{extra1} in Supplementary Figure 4. The legend of the corresponding Figures has been amended with a specific reference to the errors. For clarity the color of the profile of the calculated R_2/R_1 values for the individual subdomains has been changed to orange, to differentiate it better from the standard deviations. Details on the determination of standard deviations for R_2/R_1 and heteronuclear-NOE values has been added in Supplementary Methods dealing with relaxation measurements on p. 18 of Supplementary Information.

The R_2/R_1 values of the globular MapZ_{extra2} values in the individual subdomain and in the full-length construct overlay within experimental error and show a flat profile over the whole domain due to its isotropic shape. The R_2/R_1 values of the elongated MapZ_{extra1} values in the individual subdomain and in the full-length construct overlay reasonably well and show a similarly hieratic profile in agreement with the high anisotropy of this domain. Overall data suggests highly similar hydrodynamic properties of the two domains whether taken individually or in the full-length MapZ_{extra}.

- p.13: "Unexpectedly and strikingly..." Why is this unexpected? Wasn't the experiment designed to produce this effect?

The sentence p. 13 has been corrected and "unexpectedly and strikingly" has been deleted as indeed the experiment was designed to produce this effect, as outlined by the referee

- Maybe for the discussion: Can the serines in the linker be phosphorylated and if yes what would be the expected result?

Phosphorylation of MapZ by StkP occurs in the cytoplasm and thus concerns the two threonines that are localized in the cytoplasmic domain of MapZ (Fleurie et al, Nature, 2014). In addition, and to the best of our knowledge, there is no evidence of an extracellular kinase in the pneumococcus. The serines localized in the linker of the extracellular domain of MapZ are thus likely not phosphorylated. For clarity, the sentence mentioning that StkP phosphorylates MapZ on page 4 has been modified.

Reviewers' Comments:

Reviewer #1 (Remarks to the Author)

Reviewer #2 (Remarks to the Author)

The revised version of the manuscript should be accepted.